# A multi-sensor satellite-based archive of the largest SO₂ volcanic eruptions since 2006

Pierre-Yves Tournigand[1], Valeria Cigala[2], Elzbieta Lasota[1], Mohammed Hammouti[3], Lieven Clarisse[4], Hugues Brenot[5], Fred Prata[6], Gottfried Kirchengast[7], Andrea K. Steiner[7], Riccardo Biondi[1]

[1]Dipartimento di Geoscienze, Università degli Studi di Padova, Italy
[2]Ludwig-Maximilians-Universität München, Germany
[3]Politecnico di Milano, Italy
[4]Spectroscopy, Quantum Chemistry and Atmospheric Remote Sensing (SQUARES), Université Libre de Bruxelles, Belgium
[5]Royal Belgian Institute for Space Aeronomy (BIRA-IASB), Brussels, Belgium
[6]AIRES Pty Ltd: Mt Eliza, Victoria, Australia
[7]Wegener Center for Climate and Global Change, University of Graz, Austria

*Correspondence to:* Riccardo Biondi (riccardo@biondiriccardo.it)

**Abstract**. We present a multi-sensor archive collecting spatial and temporal information about volcanic SO₂ clouds generated by the eleven largest eruptions of this century. The detection and monitoring of volcanic clouds are an important topic for aviation management, climate issues and weather forecasts. Several studies focusing on single eruptive events exist, but no archive available at the moment combines quantitative data from as many instruments. We archived and collocated the SO₂ vertical column density estimations from three different satellite instruments (AIRS, IASI and GOME-2), atmospheric parameters as vertical profiles from the Global Navigation Satellite Systems (GNSS) Radio Occultations (RO) and the cloud top height and aerosol type from the Cloud-Aerosol LIdar with Orthogonal Polarization (CALIOP). Additionally, we provide information about the cloud top height from three different algorithms, and the atmospheric anomaly due to the presence of the cloud. The dataset is gathering 206 days of SO₂ data, collocated with 44180 backscatter profiles and 64764 radio occultation profiles. The modular structure of the archive allows an easy collocation of the datasets according to the users' needs and the cross-comparison of the datasets shows different consistency of the parameters estimated with different sensors and algorithms, according to the sensitivity and resolution of the instruments. The data described here will be published with a DOI after final acceptance of this manuscript (Tournigand et al., 2020a, http://doi.org/10.5880/fidgeo.2020.016). During the discussion period, the data are accessible via this temporary link: http://pmd.gfz-potsdam.de/panmetaworks/review/0f85d699707efcdc567765bd0dafaaadf94b6df5a531f310167f7e974ea803bf.

## 1. Introduction

Volcanoes around the world are a constant source of gaseous emissions. Both passively, during quiescent times, and actively, during eruptions (Robock, 2015; Carn et al., 2017). The most abundant gas species emitted are water ($H_2O$), carbon dioxide ($CO_2$) and sulfur dioxide ($SO_2$), while less abundant ones include hydrogen ($H_2$), hydrogen sulfur ($H_2S$), hydrochloric acid ($HCl$) and carbon monoxide ($CO$) (Williams-Jones and Rymer, 2015). Once emitted into the atmosphere, some of these gases can react and transform, for example, $SO_2$ transforms into $H_2SO_4$. As with all volcanic eruption products, the gases emitted and the related aerosols can pose hazards to people as well as the environment (Williams-Jones and Rymer, 2015). Moreover, they can be responsible for regional and global climatic effects, depending on the latitude of the volcano, the altitude reached by the eruptive column and subsequent volcanic cloud (Robock, 2000; Robock, 2015; Williams-Jones and Rymer, 2015; Carn et al., 2016). In terms of global climatic impact, $SO_2$ injections into the stratosphere are of the greatest significance. The reason is that in the stratosphere the gas and subsequent aerosol can remain suspended for months to years and hence be transported around the globe affecting the absorption of both short- and longwave radiation (Robock, 2015). The duration and spatial spreading of emitted gases and aerosols in the atmosphere also depend on the erupted mass of volcanic material and the duration of the emission.

According to the Global Volcanism Program (GVP) of the Smithsonian Institute, an average of 55 to 88 eruptions (excluding permanent and semi-permanent degassing) have occurred worldwide each year since 1994. The eruptions display variability in their eruptive style (e.g., effusive or explosive), magma composition, the energy of the eruption, and the amount, type and size of the ejected material. To compare and characterize different eruptive events in an objective manner, the Volcanic Explosivity Index (VEI) was created. The VEI was introduced in 1982 by Newhall and Self (1982), drawing inspiration from the Richter scale for earthquake's magnitude, to provide a relative, semi-quantitative measure of the explosiveness of volcanic eruptions. The VEI classification, divided into categories from 0 to 8, is based mainly on measures of magnitude, in terms of total ejecta volume, and/or intensity, in terms of mass flux or eruption plume height, depending on data availability (Newhall and Self, 1982; Houghton et al., 2013). The VEI has its limitations, nevertheless, it is being used extensively to provide an eruption descriptor that is understandable by researchers and policy-makers alike (Houghton et al., 2013).

The size of an eruption can, however, be significantly different when considering gas emissions only (Carn et al., 2016). Thus, a different eruption size classification can be outlined by the sulfur input into the atmosphere (Schnetzler et al., 1997; Robock, 2015; Carn et al., 2016). In 1997, Schnetzler et al. proposed the Volcanic Sulfur Dioxide Index (VSI) based on $SO_2$ retrievals performed with the Total Ozone Mapping Spectrometer (TOMS) onboard the Nimbus 7 satellite. Medium VEI eruptions (e.g. VEI 4 events) can be characterized by the emission of a large quantity of tephra with respect to the quantity of $SO_2$ (e.g. the 2008 Chaiten eruption) or by the emission of a larger amount of $SO_2$ than tephra (e.g. the 2011 Nabro eruption) as shown in Carn et al. (2016). Using an improved $SO_2$ emissions retrieval approach and including less energetic events, Carn et al. (2016) found a broader range of $SO_2$ emission per VEI and a weaker first-order correlation. These findings suggest that the intensity and volcanic plume altitude are more relevant parameters for consideration in modelling $SO_2$ emissions and their climate impact (Robock, 2000; Carn et al., 2016).

The above-mentioned GVP provides the most extensive catalogue of historical eruptions with information related to both volcanoes and their eruptions. This catalogue is a first-hand source of information when starting an investigation of a given volcano and a given style of eruption. Similarly, the Large Magnitude Explosive Volcanic Eruptions (LaMEVE) dataset includes data such as the magnitude of the event, the bulk volume, the tephra fallout volume, column height of Quaternary (from 2.58 Ma ago to present) eruptions with VEI $\geq$ 4 (Brown et al., 2014). Other datasets are available including data on geochemical composition (e.g. Turner et al., 2009), or acoustic acquisitions (e.g. Fee et al., 2014), although these types of database are generally limited to ancillary information, a specific volcano, a specific time window or a specific instrument (e.g. de Moor et al., 2017).

Previous studies focusing on a single or a few eruptions are based on personal data collections or project collaborations and this makes it difficult for data comparison and studies with new techniques or algorithms. The eruptions of Okmok and Kasatochi in 2008 were the focus of a special issue (JGR Atmospheres, 2018) collecting 27 articles studying the clouds with a large number of the available remote sensing platforms and algorithms. The Sarychev Peak eruption in 2009 was also well studied (e.g., Carn and Lopez, 2011; Kravitz et al., 2011; Rybin et al., 2011; Doeringer et al., 2012). The Eyjafjallajökull 2010 eruption affected the economy and social life of Europe and beyond, changing the rules of air traffic management and the volcanic cloud was subject of a number of studies (e.g., Flentje et al., 2010; Marenco et al., 2011; Prata and Prata, 2012; Stohl et al., 2012). The Grímsvötn eruption in 2011 was quite interesting from a scientific point of view because the $SO_2$ cloud was separated from the ash cloud (Moxnes et al., 2014) and different researchers studied it (e.g., Flemming and Inness, 2013; Marzano et al., 2013; Cook et al., 2014; Prata et al., 2017). The Nabro 2011 eruption was the subject of an interesting discussion regarding the direct intrusion to the stratosphere of the volcanic cloud (e.g., Bourassa et al., 2012; Clarisse et al., 2014; Fromm et al., 2014; Biondi et al., 2017) and the Puyehue Cordón Caulle, erupting in the same period, was of interest because the cloud moved around the globe (e.g., Bignami et al., 2014; Griessbach et al., 2014; Theys et al., 2014; Biondi et al., 2017). The Calbuco 2015 eruption (Marzano et al., 2018; Lopes et al., 2019) was widely studied especially in connection to its impact on the Antarctic ozone hole (Ivy et al., 2017; Stone et al., 2017; Zhu et al., 2018; Zuev et al., 2018). The rest of the volcanic clouds, such as the ones produced by Merapi 2010 (Picquout et al., 2013), Tolbachik 2012 (Telling et al., 2015), and Kelut 2014 (Kristiansen et al., 2015; Vernier et al., 2016) also received some attention from the scientific community.

Considering $SO_2$ emissions, several datasets and inventories are available and updated over time, but generally include daily or yearly total emissions per volcano or per eruption. Ge et al. (2016) compiled an inventory for daily $SO_2$ emissions in the time frame 2005-2012 including global volcanic eruptions but also eight persistently degassing volcanoes retrieved by the Ozone Monitoring Instrument (OMI) onboard the Aura satellite. Carn et al. (2017) implemented it including OMI retrievals from 2005 to 2015 of emissions related to passive degassing. The most updated and complete dataset up to now is the Multi-Satellite Volcanic Sulfur Dioxide Database Long-Term L4 Global (MSVOLSO2L4) compiled by Carn (2019). The dataset provides "$SO_2$ mass loadings for all significant global volcanic eruptions detected from space since October 1978" to present (Carn, 2019). The MSVOLSO2L4 includes ancillary information about the volcanoes, such as the name and location of the volcano, as well as information about the eruptions, for example, start and end date, and VEI. This information is retrieved from the GVP database. The dataset also reports the observed plume altitude (in kilometres) where known. Otherwise, an estimated plume altitude above vent depending on eruption type, and the measured $SO_2$ mass in kilotons (= 1000 metric tons) is provided (Carn et al., 2016; Carn, 2019). The above-mentioned datasets provide important information for users mainly needing to assess the climatic impact of $SO_2$ from volcanic sources, however, none of them allows for mapping the $SO_2$ emissions and related altitude estimations in space and time and thus the direct testing and comparison of new models and techniques, like GNSS RO, for example. We think it is important to provide a complementary multi-satellite archive covering the largest eruptive events and their cloud development all around the world in order to facilitate the access to such data for future studies.

In this study, we present the first database predicated on satellite data, reporting:

1. $SO_2$ retrievals from Atmospheric Infrared Sounder (AIRS), Infrared Atmospheric Sounding Interferometer (IASI), and the Global Ozone Monitoring Experiment 2 (GOME-2). The data from the three sensors provide horizontal and temporal information on $SO_2$ concentrations;
2. $SO_2$ cloud altitude estimations from IASI, the Cloud-Aerosol LIdar with Orthogonal Polarization (CALIOP) backscatter and the Global Navigation Satellite System (GNSS) Radio Occultation (RO);
3. the cloud aerosol subtype information from CALIOP;
4. atmospheric properties such as temperature, pressure and humidity from GNSS RO profiles.

The information is provided for eruptions, after July 2008, classified by GVP as VEI 4 or larger and with an $SO_2$ total mass loading greater than 0.05 Tg. At the time of the archive preparation no eruption after 2016 had yet been classified as VEI 4 or greater. The selected volcanoes and relative eruptions are: Okmok 2008; Kasatochi 2008; Sarychev 2009; Eyjafjallajökull 2010; Merapi 2010; Grímsvötn 2011; Nabro 2011; Puyehue-Cordón Caulle 2011; Tolbachik 2012; Kelut 2014; Calbuco 2015. These are the most significant eruptions over the period 2006-2018. Further information on these eruptions can be found in Table 1.

To the best of our knowledge, there is no database to date collecting $SO_2$ maps and volcanic cloud altitude estimations from several instruments, cloud aerosols subtypes and atmospheric properties for explosive eruptions. Accurate knowledge on volcanic $SO_2$ cloud concentration and altitude as well as their spatial and temporal evolution, is of crucial importance in the investigation of an eruption's climatic impact. Thus, we believe that the database presented here will help current and future investigations as well as support the development of more accurate retrieval methodologies.

## 2. Instrument and retrieval description

A summary of the instruments' characteristics together with parameters provided in this work and references to the algorithms are reported in Table 1. In this archive, each sensor measuring $SO_2$ amounts measures the partial column density, due to their own limitations (see section 4.3) (Brenot et al., 2014). We here use the terms Vertical Column Density (VCD) to refer to this partial column density.

### 2.1.    AIRS

The Atmospheric Infrared Sounder (AIRS) is a cross-track hyperspectral instrument onboard the polar-orbiting satellite Aqua launched in June 2002 with an ascending orbit equator crossing local time at 13:30. AIRS completely covers the full globe two times per day with a swath of 1650 km and spatial resolution of 13.5 km x 13.5 km at nadir and 41 km x 21 km at high latitudes (Susskind et al., 2003). Each orbit is divided in granules, where a granule is a portion of AIRS orbit containing 6 minutes (2250 km x 1650 km) of data, which is officially defined by the National Aeronautics and Space Administration (NASA). The $SO_2$ pixels are identified using infrared channels centered at the 7.3 μm absorption peak relying on the correlation between the measured spectrum and a reference spectral shape. The amount of $SO_2$ in each pixel is computed by a least-squares procedure based on an off-line radiative transfer model. This technique performs well for $SO_2$ reaching high tropospheric altitudes or the stratosphere where the water vapor content is negligible. Comparisons with other techniques (Carn et al., 2016; Carn et al., 2017) show an agreement within 10–30% and a typical retrieval error for a single AIRS pixel of about 6 Dobson Unit (DU) (Prata and Bernardo, 2007).

### 2.2.    IASI

The Infrared Atmospheric Sounding Interferometer (IASI) is a Fourier transform instrument onboard the near-polar sun-synchronous orbiting satellites Metop-A and Metop-B, respectively, launched in October 2006 and September 2012 with an ascending orbit equator crossing local time at 9:30. IASI completely covers the full globe two times per day with a swath of 2200 km and a spatial resolution of 12 km at nadir (Clerbaux et al., 2009). The $SO_2$ retrieval is based on a brightness temperature difference in the $SO_2$ $v_3$ band centered at 7.3 μm (Clarisse et al., 2012) which is converted to $SO_2$ concentration integrated along the vertical axis the VCD using look-up tables and operational profiles of pressure, temperature and humidity. The retrieval of VCD assumes that all $SO_2$ is located at particular atmospheric layers (5, 7, 10, 13, 16, 19, 25 or 30 km above sea level) providing different estimations at different altitudes. It has a detection limit of around 0.5 DU at the tropopause, which increases for decreasing altitude (depending on the amount of water vapour in the atmosphere). For plumes above 500 hPa (about 5.5 km) the algorithm has a theoretical uncertainty between 3-6%. A second algorithm (Clarisse et al., 2014) is applied to compute the $SO_2$ cloud altitude with an accuracy of about 2 km for plumes below 20 km. The algorithm exploits the fact that the $SO_2$ $v_3$ band interferes with strong water vapour absorptions, and that these interferences, by virtue of the vertical water vapour profile, have a strong dependency with height. Combining the two datasets, a single best-estimate VCD is obtained by interpolating the VCD columns of the first algorithm at the retrieved height.

### 2.3.    GOME

The Global Ozone Monitoring Experiment 2 (GOME-2) is an ultraviolet-visible spectrometer, onboard of the Metop-A and Metop-B satellites, measuring solar light backscattered by the atmosphere or reflected by the Earth in nadir-viewing geometry with a swath of 1920 km and spatial resolution of 40 km x 80 km at nadir (Munro et al., 2006). The $SO_2$ VCD retrieval (Rix et al., 2012) is based on the strong $SO_2$ absorption between 240 and 400 nm and uses a differential optical absorption spectroscopy technique (Platt and Stutz, 2008). All measurements in the wavelength ranging from 315 to 326 nm are fitted to laboratory absorption data of $SO_2$ and converted to VCD with an air mass factor from radiative transfer models assuming hypothetical atmospheric layers representative of different scenarios of emissions. The $SO_2$ VCD is provided for 3 atmospheric layers representative of different scenarios of emissions: low troposphere (~2.5 km above the surface), upper troposphere (~6 km) and lower stratosphere (~15 km). The volcanic emission measurement is facilitated by large $SO_2$ columns generally at high altitudes (free-troposphere to lower stratosphere). However, for large $SO_2$ columns (typically>50 DU) the absorption tends to saturate leading to a general underestimation and directly affecting the product accuracy. For most volcanoes, there is no ground-based equipment to measure $SO_2$ during the eruption and the validation approach is usually a cross-comparisons with other satellite products. The O3M SAF validation report (Theys and Koukouli, 2015) shows that GOME-2 $SO_2$ product reaches the target/optimal accuracy of 50%/30% respectively. It is important to notice that the $SO_2$ retrievals from GOME-2 are also affected by clouds and instrumental noise especially at high solar zenith angles. These limitations have been filtered in the data used in this work, according to the criteria shown by Brenot et al. (2014).

### 2.4. CALIOP

The Cloud-Aerosol LIdar with Orthogonal Polarization (CALIOP) is an instrument onboard the polar-orbiting Cloud-Aerosol Lidar and Infrared Pathfinder Satellite Observation (CALIPSO). To estimate the Volcanic Cloud (VC) top altitude and validate the cloud top estimation from GNSS RO, we have used the Level 1 total attenuated backscatter at 532 nm (CAL_LID_L1 Version 4) with a vertical resolution of 60 m and horizontal resolution of 1 km between 10 and 20 km of altitude, and a vertical resolution of 180 m and horizontal resolution of 1.67 km above 20 km (Winker et al., 2009). To extract the corresponding aerosol type we have used the Level 2 Vertical Feature Mask (Winker et al., 2009) product version 4.20. The CALIOP does not allow $SO_2$ measurements or estimation (it provides estimations of dust, elevated smoke, volcanic ash and sulfate) and the CALIOP classification algorithm do not include the volcanic ash type below the tropopause level (Kim et al. 2018) making difficult to distinguish the volcanic ash from other aerosol types in the lower troposphere. For these reasons, the selected CALIOP backscatter is collocated with the $SO_2$ estimation from AIRS, IASI and GOME-2 and this combination provides a complete information on the content and vertical structure of the cloud.

### 2.5. GNSS RO

The Global Navigation Satellite Systems (GNSS) Radio Occultation (RO) is an active limb sounding technique which uses radio signals transmitted by a GNSS satellite and received by a Low Earth Orbit satellite, where the atmosphere is vertically scanned due to the relative motion of the two satellites. The signal, travelling through the atmosphere, is refracted and bent due to the vertical gradient of atmospheric density. The effect of the atmosphere is represented by a bending angle, from which refractivity and density is retrieved. Refractivity in the neutral atmosphere depends mainly on temperature, pressure, and water vapour pressure. Information about the vertical structure of the troposphere and stratosphere is provided (Kursinski et al., 1997). The vertical resolution of the RO ranges between 100 m in the upper troposphere to about 500 m in the lower stratosphere at low/mid-latitudes (Zeng et al., 2019), while the horizontal resolution can range from about 50 km in the troposphere to 200-300 km in the stratosphere (Kursinski et al., 1997). In this archive we use the RO bending angle, refractivity, temperature, pressure and specific humidity profiles processed by the Wegener Center for Climate and Global Change (WEGC) with the Occultation Processing System (OPS) version 5.6 (EOPAC Team, 2019). We also provide the bending angle anomaly which is proven to be an efficient parameter to reveal the impact of the VC on the atmospheric structure (Biondi et al., 2017; Cigala et al., 2019; Stocker et al., 2019) because perturbations in the vertical structure are seen in the bending angle profile as anomalous peaks, specifically at the volcanic cloud top. The way this anomaly is computed is detailed in section 4.1. The RO profiles are obtained using a combination of geometric optics and wave optics retrieval (Angerer et al., 2017), with transition below the tropopause. The retrieval is based on orbit information and amplitude and phase data from the University Corporation of Atmospheric Research/COSMIC Data Analysis and Archive Center (UCAR/CDAAC) collected from the following RO missions: the CHAllenging Minisatellite Payload (CHAMP) (Wickert et al., 2001), the Satélite de Aplicaciones Científicas (SAC-C) (Hajj et al., 2004), the Gravity Recovery And Climate Experiment (GRACE-A) (Beyerle et al., 2005), the FORMOSAT-3/COSMIC (Anthes et al., 2008), and the EUMETSAT/METOP missions (Luntama et al., 2008). The accuracy of the GNSS RO is 0.2 K in terms of temperature and 0.1% in terms of refractivity and the data from the different mission are very consistent (Scherllin-Pirscher et al., 2011), so there is no need of inter-calibration or homogenization (Foelsche et al., 2011).

### 3. Data description

The archive (Tournigand et al., 2020a) consists of two sets of files, the daily files and the eruption files, i.e., one file per eruption including all collected information. Thus, for each eruption listed in Table 2 the user can choose to access one single day or to the whole eruptive period depending on the user's demand. The number of days covered by the archive for each eruption depends on the $SO_2$ detection availability from AIRS, IASI and GOME-2. Also, the variables available from one day to another may differ according to $SO_2$ detection results and instruments availability. Each file

is in NetCDF-4 format and file names are self-explanatory with daily files following the format *volcanoname_year_month_day* and the eruption files following the format *volcanoname*. As an example, a user who wishes to access all available data corresponding to the Sarychev volcano on 12 of June 2009 will have to look for the file *Sarychev_2009_06_12.nc*. The organization of both file types is described hereafter for each instrument, the main information is provided in Table 3 and all the details are provided in the supplementary material including the geographical coverage of each VC (Figures S1-S11).

### 3.1. AIRS

AIRS data are organized in the same way in both file types. It consists of 4 variables namely AIRS_lat, AIRS_lon, AIRS_date and AIRS_SO2 respectively containing the latitude (°N), longitude (°E), date and time (POSIX time, number of seconds elapsed since 00:00:00 UTC 1st of January 1970) of each granule and their $SO_2$ VCD (DU). The variables AIRS_lat, AIRS_lon and AIRS_SO2 are matrices with columns corresponding to the different granules. By selecting one column, the user can find each data point of the corresponding granule. The AIRS_date variable, on the other hand, is a line vector with elements reporting the date and time of each granule. Only data points with $SO_2$ values higher than 0 DU have been included in the archive thus explaining the different amount of points from one granule to another.

### 3.2. IASI

IASI data are organized in the same way for both file types and are composed of 5 variables, IASI_lat, IASI_lon, IASI_date, IASI_SO2 and IASI_height respectively containing the latitude (°N), longitude (°E), date and time (POSIX time), $SO_2$ (DU) and cloud altitude (m). The date variable consists of a line vector with elements corresponding to each scanning line. Similarly, the other variables are matrices with columns corresponding to the different scanning lines and rows to the data points of the given scanning line having an $SO_2$ content higher than 0 DU.

### 3.3. GOME-2

GOME-2 data's organization is identical in both file types. GOME-2 is composed of 6 variables, GOME_lat, GOME_lon, GOME_date, GOME_SO2_1, GOME_SO2_2, GOME_SO2_3 respectively corresponding to the latitude (°N), longitude (°E), date and time (POSIX time), low troposphere $SO_2$ vertical column density (DU), mid-troposphere $SO_2$ vertical column density (DU) and low stratosphere $SO_2$ vertical column density (DU). As for AIRS and IASI, the date variable corresponds to a line vector providing each scanning line's date. The rest of the variables correspond to matrices with scanning lines also separated in columns and the data points of those scanning lines distributed in rows. In the case of GOME-2, only data points having their three $SO_2$ vertical columns contents higher than 0 DU were included. Pixel with high Solar Zenith Angle (SZA) has also been filtered.

### 3.4. CALIOP

CALIOP's section of the archive contains 6 variables, CALIOP_lat, CALIOP_lon, CALIOP_date, CALIOP_filename, CALIOP_height, CALIOP_type respectively corresponding to the latitude (°N), longitude (°E), date and time (POSIX time), name of CALIOP file, estimated VC top altitude (m) and aerosol type. The latitude, longitude, date and file name variables are column vectors with each row corresponding to latitude, longitude and date data of the designated file in CALIOP_filename variable. CALIOP_height variable contains all the cloud top altitude estimations based on CALIOP L1 532 nm version 4.10 backscatter product. This variable is a matrix with each row corresponding to a CALIOP file and the three columns indicating to which instrument the CALIOP file is collocated (at ±0.2° and ±1h) with, AIRS (column 1), IASI (column 2) or GOME (column 3).

CALIOP_type variable is read as a string containing the type of aerosol retrieved from the L2 Vertical Feature Mask (VFM) version 4.20. The L2 VFM CALIOP product subdivides the aerosols into 10 types. Four of those types are of interest for this archive: type 2, 6, 9 and 10 respectively corresponding to dust, elevated smoke, volcanic ash and sulfate/other. This variable is subdivided into as many rows as there are CALIOP files, 16 columns containing the aerosols values and three sections corresponding to three levels of altitude -0.5 to 8.2 km, 8.2 to 20.2 km and 20.2 to

30.1 km. In each altitude level, the presence of one or several cloud types is indicated by the presence of their reference number.

### 3.5.    GNSS RO

GNSS RO data are organized in different ways in the two file types. In the daily files, the RO data are separated in different variables according to the instrument they are collocated with (AIRS, IASI or GOME-2) at ±0.2° spatially and ±12h temporally. For each set of RO data collocated with a given instrument 10 variables are provided: latitude (°N), longitude (°E), date (POSIX time), bending angle (rad), bending angle anomaly (%), temperature (K), pressure (Pa), refractivity (N-unit), specific humidity (kg.kg$^{-1}$) and volcanic cloud top altitude (m). Each variable is a matrix with each column corresponding to a RO profile and rows to the latitude dimension. Only the variable containing volcanic cloud altitude is a line vector with each element corresponding to a different RO profiles.

In the files containing the whole eruptive period, the RO data are not separated according to the instrument they are collocated with but compiled all together. Thus, the same 10 variables are provided as in daily files, each containing the totality of the RO profiles.

## 4.    Quality control and data processing

### 4.1.    RO data

The RO profiles included in this archive are collocated spatially at ±0.2° and temporally at ±12h with data points from the volcanic aerosol maps provided by AIRS, IASI, and GOME-2 acquisitions.

#### 4.1.1 Climatology

The RO reference climatology for each area of interest is calculated based on 5° latitude bands using our dataset of RO profiles covering a period from 2001 to 2017. The averaging of all available RO profiles present within each latitude band provides the RO reference climatology.

#### 4.1.2 Anomaly calculation

The bending angle (BA) anomaly integrated into this archive is calculated by subtracting the RO reference climatology profile from the individual RO BA profile and normalizing with respect to the reference climatology profile (Eq 1), following the methodology described in Biondi et al. (2017). The resulting anomaly displays variations when the bending angle differs from the climatology. Such variations indicate a change of atmospheric properties and are used to identify related atmospheric features.  The presence of volcanic clouds in the atmosphere generates a prominent peak in the BA anomaly profile.

$$\alpha = \left( \frac{(BA - BA_{clim})}{BA_{clim}} \right) \cdot 100 \tag{1}$$

Where α is the bending angle anomaly, BA the individual bending angle profile and BA$_{clim}$ the BA climatology profile.

#### 4.1.3 Peak detection

The peak detection of bending angle anomaly profiles was automatically done using a customized Matlab algorithm. This algorithm, further developed after Cigala et al. (2019), identifies all the peaks displaying a variation larger than 4.5% in the bending angle anomaly profile with respect to local minimums. Only the peaks having their maximum value between 10 and 22 km of altitude are kept. Peaks vertically spreading over more than 8 km have been removed. Finally, amongst the remaining peaks, the lowest altitude one is selected as a cloud top altitude.

### 4.2. CALIOP

#### 4.2.1 Cloud top automatic detection

For each L1 532 nm version 4.10 CALIOP backscatter product collocated with RO profiles, an automatic cloud top detection was performed using a customized Matlab algorithm. The collocation thresholds were kept at ±0.2° and ±12h between RO profiles and CALIOP backscatter products to provide cloud top altitudes consistent between the instruments. The first step of the cloud top detection procedure consists of cropping the CALIOP backscatter image according to the collocated RO profile position at +/-14° in latitude and +/-80° in longitude. The objective of this first

step is to focus the processing on a restricted zone around the position of the collocated RO profile in order to save computational time. These latitude and longitude ranges are based on series of tests performed on backscatter images and correspond to the best compromise between image size reduction and loss of volcanic cloud information. Threshold backscatter values are then implemented to remove the noise outside of the range $3x10^{-2}$ - $7x10^{-4}$ $km^{-1}.sr^{-1}$ to which volcanic clouds correspond. One median filter (4x3 pixels) and two Wiener filters (4x3 and 2x2 pixels) are

then successively applied to the resulting backscatter image to reduce the noise within the threshold range. Below 10 km altitude, the RO bending angle anomaly is noisy due to the presence of moisture. Thus we only included in this archive volcanic clouds with a top altitude above 10 km. The following step in the volcanic cloud top determination is then to remove image information below 10 km which also removes a significant part of meteorological clouds increasing the volcanic cloud top altitude detection accuracy. The next step of CALIOP data processing is to identify

remaining groups of pixels (or clusters) within the image. The Matlab connected components finder is set in this customized algorithm to keep only the clusters combining more than 300 pixels. Amongst these selected clusters the nine biggest ones are kept for the final processing stage. Based on all the collocations between CALIOP and the GNSS RO, a statistical analysis of volcanic clouds defined by the CALIOP cloud mask and collocated with the RO within 2 hours, has shown that the volcanic clouds are usually thinner than meteorological clouds with an aspect ratio lower

than 0.09. According to this result, the final stage of the algorithm consists of distinguishing clusters corresponding to volcanic features from the ones corresponding to meteorological clouds setting the higher limit of the aspect ratio to 0.09. Finally, the remaining clusters' top altitudes are measured and an average value calculated and saved in the archive as an estimate of the volcanic cloud top altitude.

#### 4.2.2 Cloud type detection

The cloud type detection was performed using the L2 Vertical Feature Mask (VFM) CALIOP product. These VFM products were collocated at ±0.2° and ±1h with AIRS, IASI and GOME-2 for the purpose of such data is to confirm the presence of certain aerosols types simultaneously with $SO_2$. Among the different cloud types available in VFM products the types 2, 6, 9 and 10 were of particular interest for this archive since they respectively correspond to dust, elevated smoke, volcanic ash and sulfate/other. For each CALIOP file of the archive, the aerosol subtype was extracted

using a customized Matlab routine. This Matlab algorithm reads the VFM data, detects the matching latitude/longitude points of all the CALIOP track collocated with AIRS, IASI and GOME-2 and subsets the latitude and longitude array data based on the chosen spatial window of 2°. The algorithm then extracts the feature sub-type of interest as a function of the altitude. The final output is subdivided in three levels of altitude -0.5 to 8.2 km, 8.2 to 20.2 km and 20.2 to 30.1km.

### 4.3. Uncertainties

    This archive combines five different approaches of volcanic cloud detection and each type of measurement/instrument has its own uncertainties. The comparison between different instruments, always faces uncertainties due to the spatial and temporal collocation (section 3) and to their spatial resolution (section 2). The difference in cloud top estimations can be partly explained by the different sensitivities and vertical resolution of the instruments. In addition, the number

of colocations between RO and CALIOP is much smaller than for RO-IASI and IASI-CALIOP, respectively. The cloud top height estimation for eruptions with a large number of colocations (Calbuco, Kasatochi, Nabro and Sarychev Peak) is in general consistent within the techniques. For AIRS, IASI and GOME-2 the uncertainty depends on many parameters, such as thickness of the volcanic cloud, amount of aerosols and one of the most important the unknown

volcanic cloud altitude. Thus, the error is case dependent and a general value of measurement uncertainty cannot be provided. Furthermore, the measurement noise of instruments increases over time due to instrument degradation (Lang et al., 2009; Dikty et al., 2010). However, error budgets of AIRS and IASI can be respectively found in the studies by Prata and Bernardo (2007) and Clarisse et al. (2012), while an uncertainty analysis of GOME-2 is provided by Rix et al. 2012 in the case of the 2010 Eyjafjallajökull's volcanic eruption.

## 5.    Data cross-comparison

Over the past decades, satellite data have proven efficient in volcanic cloud detection through a variety of techniques. Those data are essential in the study of the spreading of such clouds on a global scale but are scattered between the different agencies in charge of processing them. This archive gathers satellite data covering 10 VEI 4 and 1 VEI 5 eruptions from 2008 to 2016 with a total of 223 days of data coverage (Table 4).

This archive is organized in different sections (Figure 1) with each instrument estimation separated from each other. Several parameters are included within the dataset, such as $SO_2$ VCD and cloud top altitude, to allow cross-correlation between the different retrieval algorithms. The database allows the quick visualization of AIRS, IASI, GOME-2, CALIOP and RO data at a given date and time as well as the collocation of any instrument data points with another one. In order to illustrate the use of this archive, we extracted two test cases (Figure 2). The first case (Figure 2a) is the 2008 eruption of Kasatochi volcano for which we selected the 9[th] of August as reference. The second case (Figure 2b) is the 2009 Sarychev Peak eruption for which we selected the 12[th] of June as reference. In both cases we considered $SO_2$ values larger than 3 DU from AIRS, IASI and GOME-2 for 24 hours, RO profiles collocated within ±0.2° and ±12h and CALIOP tracks collocated within ±12h.
In the case of Kasatochi, we selected 4178 AIRS, 1241 IASI and 56 GOME-2 data points with $SO_2$ VCD larger than 3 DU, 379 CALIOP profiles from 7 different tracks (Figure 2a, blue circles) within ±12h and 100 RO profiles (Figure 2a, red circles) collocated within ±0.2° and ±12h. In the case of Sarychev, 1070 AIRS, 209 IASI and 41 GOME-2 data points, 261 CALIOP profiles from 3 different tracks and 54 RO profiles have been selected with the same criteria. Due to the modular archive structure reported in Figure 1, the user can easily select different time frames, different $SO_2$ thresholds and collocation period range to be adapted to any purpose.

We have manually verified the correct functioning of the algorithm which collocates the different instruments. We randomly selected several days from different eruptions and compared the date, time and coordinates of the acquisitions, then compared the results with those ones automatically provided by the algorithm. Additionally, we have used a visual validation method for all the samples plotting the $SO_2$ cloud superimposed to the CALIOP tracks and the RO tangent points. As for the cloud top height, we have collocated the RO and CALIOP estimations with the closest IASI pixel and compared the corresponding values. In Table 5 we report the number of collocations per pairs of instruments with the averaged difference between the estimation. Depending on the eruption the different techniques can give variable performances, for example, the estimations of RO and CALIOP for the Eyjafjallajökull, Kasatochi and Grímsvötn were very close (average difference of 0.3 km, 0.9 km, 1.3 km respectively) while they were large for Calbuco (4.2 km). The difference in cloud top estimations can be partly explained by the sensitivity of the RO to the density of the atmosphere, denser clouds can be detected more likely than less dense clouds (Tournigand et al., 2020b). This reason, summed up to the uncertainties reported in the section 4.3, can contribute to the different biases of the cross-comparisons. In general, the cloud top height estimation for eruptions with a large number of colocations (Calbuco, Kasatochi, Nabro and Sarychev Peak) are consistent within the techniques.

## 6.    Results

We have collected 4535062 GOME-2 scanning lines covering 182 days, 336399 IASI scanning lines covering 172 days, 865 AIRS granules covering 122 days, 44180 CALIOP profiles covering 152 days, and 64764 RO profiles covering 194 days collocated to the VC emitted by 11 different eruptions with VEI ≥4. The Kasatochi eruption has the best data coverage, Sarychev has the longest period of coverage (36 days), and Puyehue-Cordón Caulle is the only one not being covered by CALIOP (due to a technical issues on that period).

The archive allows to collocate data from five instruments working at different frequencies (Table 1), three of them (IASI, CALIOP and RO) able to provide information to develop an algorithm for the cloud top height estimation (Table 5). Part of this archive has been used in the past to develop an algorithm estimating the cloud top height by using the RO bending angle anomaly (Biondi et al., 2017; Cigala et al., 2019) and to understand the possible overshooting of the Nabro eruption in the stratosphere (Biondi et al., 2017).

The user is free to compare the $SO_2$ estimation from three different algorithms, to check the cloud structure by downloading the collocated CALIOP sub-tracks, and to analyse the impact of the volcanic cloud on the atmospheric vertical structure with the RO profiles. An example of use of this archive is shown in Figure 3. We collocated, the IASI $SO_2$ estimation the 12th of August 2008 (Figure 3e) and the AIRS $SO_2$ estimation the 11th of August 2008 (Figure 3l) of the Kasatochi VC, with the RO and CALIOP profiles. The map visually provides the $SO_2$ distribution and the position of the RO and CALIOP profiles. Panels d) and i) show the vertical distribution of the aerosol according to the CALIOP algorithm: on the 12th of August there was volcanic ash together with sulphate up to 13 km of altitude, while on the 11th of August the sulphate was prevailing and the cloud top was slightly lower (about 12 km). We then report the vertical profiles of temperature anomaly (c and h) and water vapour anomaly (a and f) when compared to the climatological values of the area. The behaviour of the two parameters are similar, but on the 12th of August the temperature anomaly in the lower troposphere is colder, and the water vapour anomalies are larger in the lower troposphere and at the cloud top layer. Finally, the bending angle anomaly (b and g) according to the algorithm reported in section 4.1.3, shows a cloud top height of 11.2 km (the 12th of August) against about 13 km from CALIOP, and a cloud top height of 11.9 km (the 11th of August) slightly lower than the detection from CALIOP. This comparison shows that the cloud structure was steady over the time with similar characteristics confirmed by four different instruments.

## 7. Data availability

The raw CALIOP data can be found at https://urs.earthdata.nasa.gov/.
The archive consists of daily files and "eruption" files. For each eruption, we provide access to single daily files or to one file for the whole eruptive period. The files of any eruption are compressed (.zip) NetCDF-4 format (including the daily and whole eruptive period) together with two pdfs (Supplement) describing the file structure. The file names are self-explanatory with daily files following the format *volcanoname_year_month_day.nc* and the eruption files following the format *volcanoname.nc*. As an example, a user who wishes to access the data corresponding to Kasatochi volcano on 11th of August 2008 will have to look for the file *Kasatochi_2008_08_11.nc*. In case the user wants all the available data for the Kasatochi eruption, they will have to look for the file *Kasatochi.nc*. The data structure of daily files and volcano files is reported in the Supplement. The data described here will be published with a DOI after final acceptance of this manuscript (Tournigand et al., 2020a, http://doi.org/10.5880/fidgeo.2020.016). During the discussion period, the data are accessible via this temporary link: http://pmd.gfz-potsdam.de/panmetaworks/review/0f85d699707efcdc567765bd0dafaaadf94b6df5a531f310167f7e974ea803bf.

## 8. Summary and conclusions

This paper presents the first comprehensive archive with quantitative information on large $SO_2$ volcanic clouds since 2006. We collected three different datasets of volcanic $SO_2$ detection from AIRS, IASI and GOME-2 instruments and co-located the detected pixels with the CALIOP and the GNSS RO products to get information about the cloud vertical structure. The archive provides information about the $SO_2$ detection and retrieval (with 3 different algorithms), the cloud top height (with 3 different algorithms), the cloud aerosol type (CALIOP vertical mask feature reference), the atmospheric parameters (bending angle, refractivity, temperature, pressure and specific humidity) and the atmospheric change due to the presence of the volcanic cloud (bending angle anomaly). At present, there are no public archives of volcanic clouds which can be used as a reference for further studies and all the information is scattered in different locations and available under different conditions. The aim of this archive and this paper is to provide the users with a complete set of state-of-the-art data. The interest in volcanic clouds detection and monitoring is high and there are still some challenges like the accurate determination of the cloud top height and cloud density to be faced. This archive

will make available to the scientific community a relevant number of cases to develop and test new algorithms on, thereby contributing to improving the accuracy on the estimation of fundamental volcanic clouds parameters. The modular structure of the archive can be easily extended in the future to smaller eruptions (VEI<4) and to other $SO_2$ estimations, facilitating the inter/cross-comparison between algorithms, allowing the reconstruction of the cloud structure and dynamical characteristics and supporting the development of cloud dispersion models.

**Author contributions.** PYT and VC collected the data. LC provided the IASI dataset. HB provided the GOME-2 dataset. FP provided the AIRS dataset. GK and AS provided the WEGC GNSS RO dataset. PYT, VC and EL conceived the algorithm approach and wrote the code. MH elaborated the CALIOP VFM data. PYT, VC and RB structured and wrote the manuscript. RB conceived the idea, coordinated the team, supervised the project and acquired the funding. EL, MH, LC, HB, FP, GK and AS reviewed the manuscript.

**Acknowledgement.** The work is accomplished in the frame of the VESUVIO project funded by the Supporting Talent in ReSearch (STARS) grant at Università degli Studi di Padova, IT. We would like to thank Armin Leuprecht for the support and all the suggestions to make the archive technically correct.

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

**Table 1. Summary of data used to build the archive.**

| Sensor | Satellite(s) | Vertical resolution | Spatial resolution | Estimation | Wavelength | Algorithm reference |
|--------|-------------|--------------------|--------------------|-----------|-----------|--------------------|
| *AIRS* | Aqua (A-Train) | NA | 13.5 km x 13.5 km | $SO_2$ VCD | 7.3 μm | Prata and Bernardo, 2007 |
| *IASI* | MetOp-A/B | NA | 12 km diameter | $SO_2$ VCD | | Clarisse et al., 2012 |

| | | | | Cloud top height | 3.62 - 15.5 µm | Clarisse et al., 2014 |
|---|---|---|---|---|---|---|
| *GOME-2* | MetOp-A/B | NA | 40 km x80 km | SO₂ VCD | 240 - 400 nm | Rix et al., 2012 |
| *CALIOP* | CALIPSO | 60 m below 20 km <br> 180 m above 20 km | 1 km <br><br> 1.667 km | Cloud top height <br> Cloud type | 532 nm <br> 1064 nm | Winker et al., 2009 |
| *GNSS RO* | CHAMP COSMIC C/NOFS SAC-C GRACE-A Met-Op | 100 m in the troposphere <br> 600 m in the stratosphere | 50 km in the troposphere <br> 200–300 km in the stratosphere | Bending angle <br> Bangle anomaly <br> Refractivity <br> Temperature <br> Pressure <br> Specific Humidity <br> Cloud top height | $19.05*10^4$ µm <br> $24.40*10^4$ µm | Angerer et al., 2017 <br> Cigala et al., 2019 |

**Table 2. Summary of the volcanoes and related eruption selected for the database. The following information is provided:**
**the name of each volcano; the eruption start date as provided by the GVP; the spatial location of the volcano in latitude and longitude; the plume/cloud height as a range estimated from IASI, CALIOP and GNSS RO (see details below) and the SO₂ mass loading in Tg as reported in the literature.**

| Volcano name | Main eruptive event date | VEI | Location Latitude/Longitude (degree) | Archive start/end date | Cloud average height (km) - Sensor | SO₂ mass loading (Tg) (Reference) Sensor |
|---|---|---|---|---|---|---|
| Okmok (Figure S1) | 12.07.2008 | 4 | 53.397/-168.166 | 12.07.2008 06.08.2008 | 12.6 - IASI <br> 12.0 - CALIOP <br> 12.5 - RO | 0.12 (Spinei et al., 2010) OMI <br> 0.3 (Prata et al. 2010) AIRS <br> 0.09 (Carn et al., 2016) IASI |
| Kasatochi (Figure S2) | 07.08.2008 | 4 | 52.172/-175.509 | 07.08.2008 29.08.2008 | 11.7 - IASI <br> 12.0 - CALIOP <br> 12.4 - RO | 2.7 (Corradini et al., 2010) MODIS <br> 1.2 (Prata et al., 2010) AIRS <br> 2.2 (Krotkov et al., 2010) OMI <br> 2.0 (Yang et al., 2010) OMI <br> 1.7 (Karagulian et al.) IASI <br> 1.7 (Kristiansen et al., 2010) Multi-sensor <br> 1.6 (Clarisse et al., 2012) IASI <br> 1.6 (Nowlan et al., 2011) GOME-2 |
| Sarychev (Figure S3) | 14.06.2009 | 4 | 48.092/153.200 | 11.06.2009 16.07.2009 | 12.2 - IASI <br> 12.9 - CALIOP <br> 12.3 - RO | 1.2 (Haywood et al., 2010) IASI |
| Eyjafjallajökull (Figure S4) | 20.03.2010 | 4 | 63.633/19.633 | 05.05.2010 21.05.2010 | 8.0 - IASI <br> 12.2 - CALIOP <br> 12.3 - RO | 0.17 (Boichu et al., 2013) IASI <br> 1.2 (Rix et al., 2012) GOME-2 <br> 0.18 (Carboni et al., 2012) IASI <br> 0.06 (Pugnaghi et al., 2016) MODIS |

| | | | | | | |
|---|---|---|---|---|---|---|
| Merapi (Figure S5) | 04.11.2010 | 4 | -7.542/110.442 | 26.10.2010 11.11.2010 | 12.4 - IASI 14.5 - CALIOP 16.1 - RO | 0.44 (Surono et al., 2012) Multi-sensor |
| Grímsvötn (Figure S6) | 21.05.2011 | 4 | 64.416/-17.316 | 22.05.2011 18.06.2011 | 10.8 - IASI 12.2 - CALIOP 11.7 - RO | 0.24 (Prata et al., 2017) 0.38 (Carn et al., 2016) 0.4 (Sigmarsson et al., 2013) OMI+IASI 0.61 (Moxnes et al., 2014) |
| Nabro (Figure S7) | 12.06.2011 | 4 | 13.370/41.700 | 31.05.2011 25.06.2011 | 12.2 - IASI 14.3 - CALIOP 15.3 - RO | 4.5 (Theys et al., 2013) Multi-sensor |
| Puyehue-Cordón Caulle (Figure S8) | 04.06.2011 | 5 | -40.59/-72.117 | 07.06.2011 18.06.2011 | 12.2 - IASI NA - CALIOP 12.5 - RO | 0.2 (Theys et al., 2013) IASI |
| Tolbachik (Figure S9) | 27.11.2012 | 4 | 55.832/160.326 | 27.11.2012 03.12.2012 | 8.9 - IASI 11.4 - CALIOP 11.7 - RO | 0.2 (Telling et al. 2015) Multi-sensor 0.09 (Carn et al., 2016) IASI |
| Kelut (Figure S10) | 13.02.2014 | 4 | -7.939/112.307 | 17.02.2014 18.02.2014 | 17.6 - IASI NA - CALIOP 16.9 - RO | 0.2 (Carn et al., 2016) OMI 0,19 (Carn et al., 2016) IASI |
| Calbuco (Figure S11) | 22.04.2015 | 4 | -41.328/-72.607 | 24.04.2015 24.05.2015 | 16.4 - IASI 12.4 - CALIOP 14.8 - RO | 0.3 (Pardini et al., 2018) |

**Table 3. General description of all the variables contained in the archive**

| Variable name | Content | Dimension (rows, columns) | Type | Unit |
|---|---|---|---|---|
| AIRS_lat | Latitude data, each column corresponds to a granule and each row to one data point in a granule. | AIRS_lat, date_AIRS | double | degrees north |
| AIRS_lon | Longitude data, each column corresponds to a granule and each row to one data point in a granule. | AIRS_lat, date_AIRS | double | degrees east |
| AIRS_date | Date of granule contained in each column. | 1, date_AIRS | int | seconds since 1970-01-01 00:00:0.0 |
| AIRS_SO2 | SO2 data, each column corresponds to a granule and each row to one data point in a granule. | AIRS_lat, date_AIRS | double | DU |
| IASI_lat | Latitude data, each column corresponds to a granule and each row to one data point in a granule. | IASI_lat, date_IASI | double | degrees north |
| IASI_lon | Longitude data, each column corresponds to a granule and each row to one data point in a granule. | IASI_lat, date_IASI | double | degrees east |
| IASI_date | Date of granule contained in each column. | 1, date_IASI | int | seconds since 1970-01-01 00:00:0.0 |

| IASI_SO2 | SO2 data, each column corresponds to a granule and each row to one data point in a granule. | IASI_lat, date_IASI | double | DU |
|---|---|---|---|---|
| IASI_height | Cloud top height estimated with IASI | IASI_lat, date_IASI | double | m |
| GOME_lat | Latitude data, each column corresponds to a granule and each row to one data point in a granule. | GOME_lat, date_GOME | double | degrees north |
| GOME_lon | Longitude data, each column corresponds to a granule and each row to one data point in a granule. | GOME_lat, date_GOME | double | degrees east |
| GOME_date | Date of granule contained in each column. | 1, date_GOME | int | seconds since 1970-01-01 00:00:0.0 |
| GOME_SO2_1 | SO2 data, each column corresponds to a granule and each row to one data point in a granule. | GOME_lat, date_GOME | double | DU |
| GOME_SO2_2 | SO2 data, each column corresponds to a granule and each row to one data point in a granule. | GOME_lat, date_GOME | double | DU |
| GOME_SO2_3 | SO2 data, each column corresponds to a granule and each row to one data point in a granule. | GOME_lat, date_GOME | double | DU |
| CALIOP_lat | Latitude data, each row corresponds to one point of a CALIOP track. | CALIOP_lat, 1 | double | degrees north |
| CALIOP_lon | Longitude data, each row corresponds to one point of a CALIOP track. | CALIOP_lat, 1 | double | degrees east |
| CALIOP_date | Date and time, each row corresponds to one point of a CALIOP track. | CALIOP_lat, 1 | int | seconds since 1970-01-01 00:00:0.0 |
| CALIOP_filename | Filename, each row provides the filename of the given data point. | CALIOP_lat, CALIOP_char | char | n.a. |
| CALIOP_height | Cloud top altitude data, each row corresponds to one point of a CALIOP track and each column to a collocated sensor. | CALIOP_lat, Sensors | double | m |
| CALIOP_type | Cloud type data, each row corresponds to one point of a CALIOP track, three columns corresponding to three levels of altitude -0.5 to 8.2 km, 8.2 to 20.2km and 20.2 to 30.1km | CALIOP_lat, CALIOP_char2, CALIOP_type | double | n.a. |
| **Only volcano files** | | | | |
| RO_lat | Latitude data, each row corresponds to one profile point and each column to a ro profile. | RO_lat, RO_profile | double | degrees north |
| RO_lon | Longitude data, each row corresponds to one profile point and each column to a ro profile. | RO_lat, RO_profile | double | degrees east |
| RO_date | Date and time data, each row corresponds to one profile point and each column to a ro profile. | RO_lat, RO_profile | int | seconds since 1970-01-01 00:00:0.0 |
| RO_bending_angle | Bending angle data, each row corresponds to one profile point and each column to a ro profile. | RO_lat, RO_profile | double | rad |
| RO_anomaly_bending_angle | Bending angle anomaly data, each row corresponds to one profile point and each column to a ro profile. | RO_lat, RO_profile | double | percent |
| RO_temperature | Temperature data, each row corresponds to one profile point and each column to a ro profile. | RO_lat, RO_profile | double | K |
| RO_pressure | Pressure data, each row corresponds to one profile point and each column to a ro profile. | RO_lat, RO_profile | double | Pa |
| RO_refractivity | Refractivity data, each row corresponds to one profile point and each column to a ro profile. | RO_lat, RO_profile | double | 1 |
| RO_specific_humidity | Specific humidity data, each row corresponds to one profile point and each column to a ro profile. | RO_lat, RO_profile | double | kg.kg-1 |
| RO_heightVC | Cloud top altitude data, each column corresponds to a ro profile. | 1, RO_profile | double | m |
| **Only daily files** | | | | |
| RO_AIRS_lat | Latitude data, each row corresponds to one profile point and each column to a ro profile. | RO_AIRS_lat, RO_AIRS_profile | double | degrees north |

| RO_AIRS_lon | Longitude data, each row corresponds to one profile point and each column to a ro profile. | RO_AIRS_lat, RO_AIRS_profile | double | degrees east |
|---|---|---|---|---|
| RO_AIRS_date | Date and time data, each row corresponds to one profile point and each column to a ro profile. | RO_AIRS_lat, RO_AIRS_profile | int | seconds since 1970-01-01 00:00:0.0 |
| RO_AIRS_bending_angle | Bending angle data, each row corresponds to one profile point and each column to a ro profile. | RO_AIRS_lat, RO_AIRS_profile | double | rad |
| RO_AIRS_anomaly_bending_angle | Bending angle anomaly data, each row corresponds to one profile point and each column to a ro profile. | RO_AIRS_lat, RO_AIRS_profile | double | percent |
| RO_AIRS_temperature | Temperature data, each row corresponds to one profile point and each column to a ro profile. | RO_AIRS_lat, RO_AIRS_profile | double | K |
| RO_AIRS_pressure | Pressure data, each row corresponds to one profile point and each column to a ro profile. | RO_AIRS_lat, RO_AIRS_profile | double | Pa |
| RO_AIRS_refractivity | Refractivity data, each row corresponds to one profile point and each column to a ro profile. | RO_AIRS_lat, RO_AIRS_profile | double | 1 |
| RO_AIRS_specific_humidity | Specific humidity data, each row corresponds to one profile point and each column to a ro profile. | RO_AIRS_lat, RO_AIRS_profile | double | kg.kg-1 |
| RO_AIRS_heightVC | Cloud top altitude data, each column corresponds to a ro profile. | 1, RO_AIRS_profile | double | m |
| RO_IASI_lat | Latitude data, each row corresponds to one profile point and each column to a ro profile. | RO_IASI_lat, RO_IASI_profile | double | degrees north |
| RO_IASI_lon | Longitude data, each row corresponds to one profile point and each column to a ro profile. | RO_IASI_lat, RO_IASI_profile | double | degrees east |
| RO_IASI_date | Date and time data, each row corresponds to one profile point and each column to a ro profile. | RO_IASI_lat, RO_IASI_profile | int | seconds since 1970-01-01 00:00:0.0 |
| RO_IASI_bending_angle | Bending angle data, each row corresponds to one profile point and each column to a ro profile. | RO_IASI_lat, RO_IASI_profile | double | rad |
| RO_IASI_anomaly_bending_angle | Bending angle anomaly data, each row corresponds to one profile point and each column to a ro profile. | RO_IASI_lat, RO_IASI_profile | double | percent |
| RO_IASI_temperature | Temperature data, each row corresponds to one profile point and each column to a ro profile. | RO_IASI_lat, RO_IASI_profile | double | K |
| RO_IASI_pressure | Pressure data, each row corresponds to one profile point and each column to a ro profile. | RO_IASI_lat, RO_IASI_profile | double | Pa |
| RO_IASI_refractivity | Refractivity data, each row corresponds to one profile point and each column to a ro profile. | RO_IASI_lat, RO_IASI_profile | double | 1 |
| RO_IASI_specific_humidity | Specific humidity data, each row corresponds to one profile point and each column to a ro profile. | RO_IASI_lat, RO_IASI_profile | double | kg.kg-1 |
| RO_IASI_heightVC | Cloud top altitude data, each column corresponds to a ro profile. | 1, RO_IASI_profile | double | m |
| RO_GOME_lat | Latitude data, each row corresponds to one profile point and each column to a ro profile. | RO_GOME_lat, RO_GOME_profile | double | degrees north |
| RO_GOME_lon | Longitude data, each row corresponds to one profile point and each column to a ro profile. | RO_GOME_lat, RO_GOME_profile | double | degrees east |
| RO_GOME_date | Date and time data, each row corresponds to one profile point and each column to a ro profile. | RO_GOME_lat, RO_GOME_profile | int | seconds since 1970-01-01 00:00:0.0 |
| RO_GOME_bending_angle | Bending angle data, each row corresponds to one profile point and each column to a ro profile. | RO_GOME_lat, RO_GOME_profile | double | rad |
| RO_GOME_anomaly_bending_angle | Bending angle anomaly data, each row corresponds to one profile point and each column to a ro profile. | RO_GOME_lat, RO_GOME_profile | double | percent |
| RO_GOME_temperature | Temperature data, each row corresponds to one profile point and each column to a ro profile. | RO_GOME_lat, RO_GOME_profile | double | K |
| RO_GOME_pressure | Pressure data, each row corresponds to one profile point and each column to a ro profile. | RO_GOME_lat, RO_GOME_profile | double | Pa |
| RO_GOME_refractivity | Refractivity data, each row corresponds to one profile point and each column to a ro profile. | RO_GOME_lat, RO_GOME_profile | double | 1 |

| | | | | |
|---|---|---|---|---|
| RO_GOME_specific_humidity | Specific humidity data, each row corresponds to one profile point and each column to a ro profile. | RO_GOME_lat, RO_GOME_profile | double | kg.kg-1 |
| RO_GOME_heightVC | Cloud top altitude data, each column corresponds to a ro profile. | 1, RO_GOME_profile | double | m |
| RO_AIRS_lat | Latitude data, each row corresponds to one profile point and each column to a ro profile. | RO_AIRS_lat, RO_AIRS_profile | double | degrees north |

**Table 4. Number of days, granules, scanning lines and profiles covered by the archive for each volcano in alphabetical order**

| Volcano | CALIOP | | AIRS | | IASI | | GOME | | RO | |
|---|---|---|---|---|---|---|---|---|---|---|
| | # of profiles | # of days covered | # of granules | # of days covered | # of scanning lines | # of days covered | # of scanning lines | # of days covered | # of profiles | # of days covered |
| Calbuco | 12495 | 30 | 350 | 30 | 42740 | 30 | 20992 | 5 | 5362 | 31 |
| Eyjafjallajökull | 3569 | 16 | 76 | 16 | 3980 | 16 | 164369 | 16 | 2624 | 17 |
| Grímsvötn | 6268 | 21 | 147 | 8 | 49824 | 20 | 833541 | 21 | 6007 | 21 |
| Kasatochi | 12897 | 23 | 247 | 13 | 103622 | 21 | 650031 | 23 | 17045 | 23 |
| Kelut | 72 | 2 | 1 | 1 | 1313 | 1 | 2575 | 1 | 83 | 2 |
| Merapi | 1053 | 11 | 27 | 10 | 4919 | 15 | 80193 | 16 | 984 | 17 |
| Nabro | 2463 | 11 | 123 | 12 | 59359 | 11 | 638316 | 9 | 7131 | 14 |
| Okmok | 5678 | 23 | 32 | 11 | 2931 | 18 | 737981 | 26 | 13255 | 26 |
| PCC | 0 | 0 | 76 | 11 | 21528 | 11 | 369992 | 11 | 664 | 12 |
| Sarychev | 11563 | 34 | 127 | 17 | 83533 | 35 | 1035931 | 36 | 16522 | 36 |
| Tolbachik | 617 | 7 | 9 | 2 | 5390 | 5 | 22133 | 5 | 449 | 7 |

**Table 5. The average difference between cloud top height estimated with pairs of sensors for each volcano. For each pair we report the average difference (or simply the difference when there is only 1 collocation) of all the collocations and the number of collocations. When there are no collocations the value reported is "/".**

| Volcano | RO-CALIOP altitude average (km) | # | RO-IASI altitude average (km) | # | IASI-CALIOP altitude average (km) | # |
|---|---|---|---|---|---|---|
| Calbuco | 4.2 | 39 | 3.4 | 867 | 4.9 | 308 |
| Eyjafjallajökull | 0.3 | 1 | 3.7 | 30 | 3.5 | 29 |
| Grímsvötn | 0.9 | 5 | 3.7 | 136 | 2.4 | 75 |
| Kasatochi | 1.3 | 70 | 1.2 | 3855 | 1.6 | 997 |
| Kelut | / | 0 | 1.7 | 20 | / | 0 |
| Merapi | 1.5 | 1 | 2.7 | 127 | 2.2 | 70 |
| Nabro | 3.4 | 9 | 4.3 | 609 | 3.6 | 204 |
| Okmok | 3.3 | 2 | 1.8 | 143 | 2.5 | 22 |
| Puyehue-Cordón Caulle | / | 0 | 1.6 | 193 | / | 0 |
| Sarychev | 1.5 | 24 | 1.5 | 1519 | 2.8 | 227 |
| Tolbachik | / | 0 | 3.0 | 68 | / | 0 |

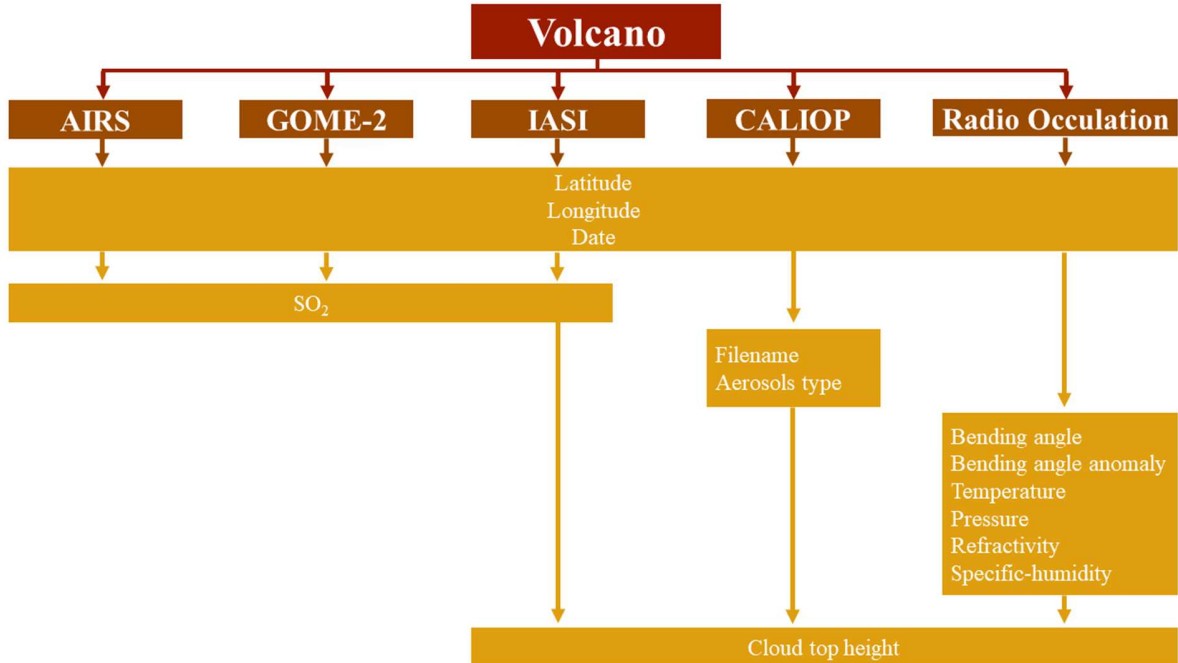

**Figure 1. Archive schematic organization.**

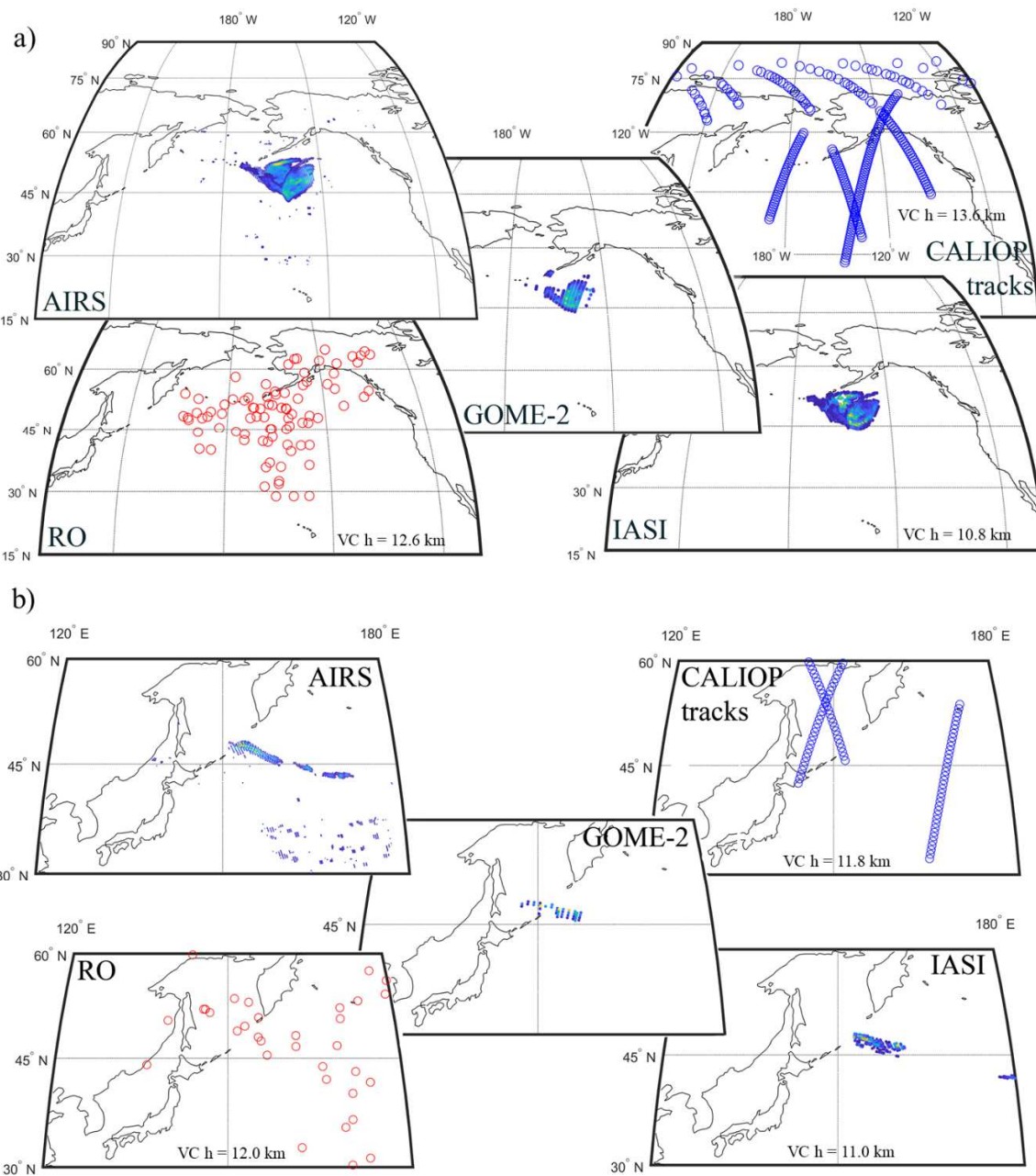

**Figure 2. Example of data use and data collocation. (a) Kasatochi cloud on 9[th] of August 2008; (b) Sarychev Peak cloud on 12[th] of June 2009. The central panels show the SO₂ VCD 2-dimensional spreading estimated by AIRS, GOME-2 and IASI, the top right panel show the CALIOP tracks for which the total attenuated backscatter profile is available and the bottom-**

710 **left panel shows the RO profiles collocated with the SO₂. For IASI, CALIOP and RO the average volcanic cloud top altitude for the considered day is indicated (VC h).**

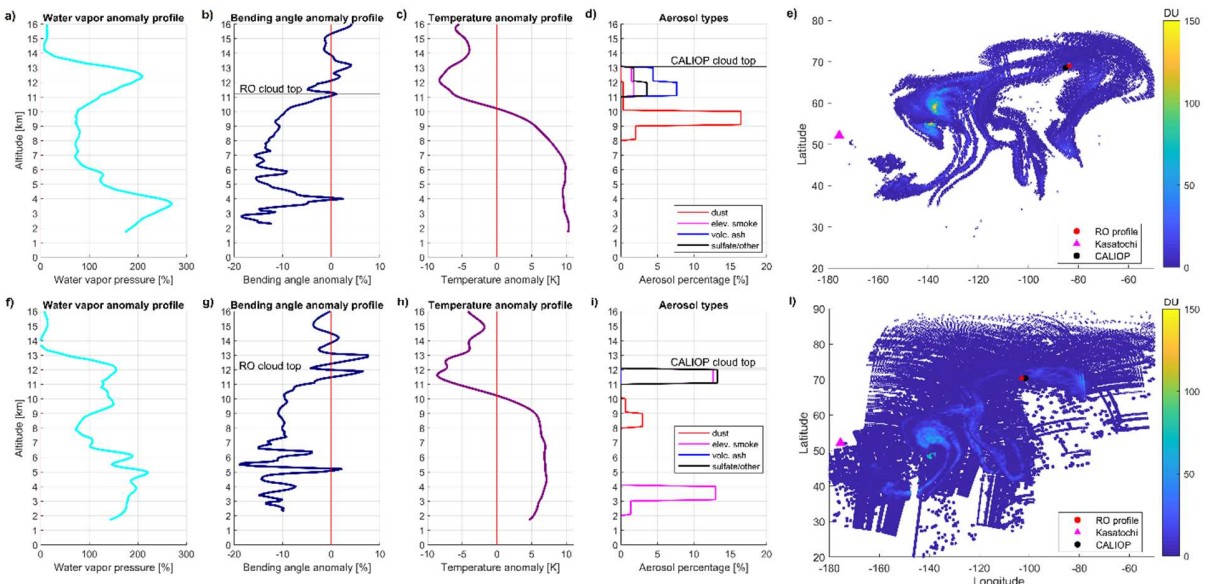

**Figure 3. Case study Kastochi 2008. RO profiles corresponding to the CALIOP aerosol types profile collocated with the SO₂ estimation from IASI (top panels, 12/08/2008) and AIRS (bottom panels, 11/08/2008).**