# Peer review of "A multi-sensor satellite-based archive of the largest SO2 volcanic eruptions since 2006"

_Earth System Science Data, 2020_

## Referee Comment (RC1) · Anonymous Referee #1 · 27 Aug 2020

The submitted manuscript describes a dataset that brings together information about SO2 emissions from 11 of the largest eruptions which have occurred since 2006. This dataset collates the results from multiple satellite instruments making it a very valuable resource for future studies of these eruptions and for the validation of new tools for monitoring plumes of SO2 from volcanoes. While the retrieval schemes are not themselves new, this is the first effort to collate this type of data in one location and one simple format. The data was easily accessible and simple to download. I read and plotted the data for a number of eruptions from the dataset in IDL using an existing NetCDF reader and using the manuscript and supplementary information for guidance. For the most part I found this to be straightforward. Having said that, further work is

required to improve the clarity of the paper so that the data can be easily read and interpreted by the users. In particular, further thorough proof reading is encouraged to improve the readability and there should be more discussion of the uncertainties and strengths/limitations of each tool so that someone using the data interprets it correctly. There are also some inconsistencies in the dataset that should be improved. See below for the specific and technical changes suggested for the manuscript and the dataset.

Specific Comments

MANUSCRIPT

Line 17. I feel it would be beneficial to elaborate here on there being no other archive which compiles the results from multiple satellite instruments and eruptions to really emphasise the strength of this dataset.

Line 20. 'We've archived and collocated ... the vertical backscatter from CALIOP...' – I think rather than 'vertical backscatter' this sentence should indicate that you have included the CALIOP height and aerosol type.

Line 25. Here you state that 'the cross-comparison of the datasets shows the high consistency of the parameters estimated with different sensors and algorithms'. This feels like quite a strong statement. In section 5 you compare the heights obtained with RO, CALIOP and IASI. You note that for a number of eruptions there is a good agreement between RO and CALIOP but that this is not the case for Calbuco. Table 4 shows that a number of the average differences between IASI/CALIOP and IASI/RO are greater than 3 km. Additionally, you have done no quantitative comparison of the partial column densities from AIRS, IASI and GOME-2. Some rewording would improve this statement.

Line 91-102. This paragraph is a little confusing. You cite three papers/datasets: Ge et al. (2016); Carn et al. (2017); Carn et al. (2019) and it is a little difficult to tell if and

how these papers/datasets are connected. Also, you suggest that Carn et al. (2017) included 'passive degassing' and 'main eruptive events' (line 94) but to the best of my knowledge this paper generates long term averaged fluxes that exclude large eruptive events. I would advise some rewording of this paragraph and perhaps some expansion on what is included in the Ge et al. (2016) and Carn et al. (2017) papers which might to help the reader better understand their content.

Section 2. In this section it would be useful to have some more information about the performance of each technique. For example, conditions in which the technique performs well or badly. And information such as the detection limits and uncertainties. This has been done for AIRS (lines 135-138) and something similar for each instrument/technique would help the reader appreciate the strengths and limitations of each tool. It would also help a user to correctly interpret the archived data- especially if they are comparing the results from different instruments. Section 4.3 does point the reader to some of the relevant literature but it would be nice to have this in section 2 and with more detail.

Line 144. You mention the IASI retrieval technique is based on a BTD with the v3 absorption band – brightness temperature difference with what?

Line 144-149. Initially it is implied that the IASI VCD retrieval is run using fixed heights. But in the archived data there is only a single value for the IASI VCD. Could you clarify if this is obtained by interpolating the results with the height from the second retrieval?

Line 148. It would be useful if there was a line here explaining how the IASI height retrieval worked.

Section 2.4. I think in this section it is important to highlight that with CALIOP you are not measuring SO2 but ash, sulphate, smoke and/or dust. It would be good to acknowledge here some of the limitations of assuming SO2/ash are collocated.

Section 3. This section details all the variables contained in the files. I think it would be

really beneficial to a user to have these listed in a table (either in this paper or in the supplementary information). I found I referred to the supplementary information (print out of all the variables) a lot while trying to load and plot the data. A table summarising the variable names, meaning, dimensions, type and units would be even more useful as a quick reference guide.

Line 201. This is the first instance that 'granule' has been used. Please can this be clearly defined. The use of the word granule made it challenging to interpret the data structures described in sections 3.1-3.3 independently of reading in and looking at the data.

Section 3.4, Section 3.5. In these sections you mention collocation with between CALIOP, RO and the other instruments. I think it would be useful to know what conditions you use for the collocation here rather than in sections 4.1 and 4.2. This would help the reader immediately understand what is meant by collocation.

Lines 340-344. Here you discuss the average differences between the cloud heights for different eruptions. Are there any reasons why the average difference is greater at Calbuco than for Eyja, Kasatochi and Grimsvotn? Also, is there a reason why the differences are greater between IASI and RO/CALIOP? I think it would be useful for a user of the dataset to understand why differences might arise between the different datasets (e.g. the time difference between the overpasses and the method used to obtain the height information). Additionally, have you considered a quantitative comparison of the VCDs retrieved with AIRS, IASI and GOME-2? What differences would you expect to see between these?

Table 1. You could add the eruption VEI and the eruption end date or duration to this table. Additionally, it could be helpful to add the geographic region considered for each eruption and the start/end date for the data in the archive – both of these would be valuable to the data user.

Table 2. In addition to the information given you could also mention the spectral
range/resolution of the instruments.

Figure 2. It would be interesting to see the cloud top heights obtained with CALIOP and RO in this plot rather than just the tracks/points.

SUPPLEMENT AND DATA

NULL values – Throughout much of the dataset the null values are reported as -9999. However, for the RO profiles they are recorded as NaN. For the RO cloud top heights it goes back to -9999. For the CALIOP heights there are no -9999 or NaN instead there are 0's- are these null values too? This should probably be consistent and whichever is chosen should be clearly noted somewhere.

Different number of variables - The files do not contain consistent numbers of variables. For example, in the file 'Calbuco_2015_05_24.nc' there is data available for IASI and RO but not GOME-2, AIRS or CALIOP. Presumably this is related to the availability of the data. It would be good to clarify this in the manuscript (perhaps at the start of section 3). Even better would be to summarise how many days or which days are covered by each instrument for each eruption – this could be an addition to table 3 and would be slightly easier to interpret than the number of granules.

Dimensions - a list of dimensions is given on page 1 and page 8 of the supplement. It would be very helpful if these were expanded on. In particular the definitions for 'CALIOP_char' and 'CALIOP_char2' are not very informative.

P1, P2 supplementary info – there is a slight discrepancy between the long names between IASI and GOME-2. For GOME-2 the long name states that the data is a composite of GOME-A and B – is it also the case for IASI that the data is a composite of IASI-A and B?

IASI_SO2 – I suggest expanding the long name of this variable to make it clear that this is a vertical column density and to explain what interpolated is referring to

P3, P4. It is not clear what the dimensions should be here. CALIOP_CHAR,

[Figure]

CALIOP_char2 and CALIOP_type should be more clearly defined in the supplementary data. CALIOP_type (the dimension) is not defined in the dimensions list.

CALIOP_type – This variable was very challenging for me to read in (in both IDL and python). The supplementary information (page 4) suggested that these were doubles but they had to be read in as strings. I think the choice of saving these as a string is so that multiple flags can be indicated. Initially on reading in this variable I obtained an array with 3 dimensions. These then had to be converted to strings and joined together to extract the CALIOP type (a similar thing had to be done for CALIOP_filename- also not immediately obvious how to read in IDL). Following that the newly joined strings had to be searched to determine which aerosols were present. Could there be a better way of saving this variable? Perhaps simply an integer array for each variable type with 1 indicating the presence of this aerosol and 0 indicating its absence. Alternatively, more information on how to read in and interpret these results would be very useful.

P5-P7. For the RO variables expanding the long names for 'air_temperature', 'air_pressure', 'refractivity', 'specific_humidity' would provide more information- these could for example mention that these are profiles.

RO – cloud top heights. The units do not seem to be consistent for these (in the daily files). For colocations with AIRS and IASI the heights appear to be in meters (which are the standard units and consistent with heights reported by CALOP and IASI). Whereas for GOME-2 they seem to be in km.

P4-7. The dimensions for the RO profiles are listed as RO_AIRS_lat by RO_AIRS_PROFILE (or IASI/GOME). Could these be defined more clearly in the dimensions list.

Dates covered by each eruption. Some of the daily files start before the start date of the eruption. For example, for Nabro (eruption starting on the 13th June 2011) the first file in the dataset is 31st May 2011. In the first few files it seems to include the outputs for other eruptions. For example, the file Nabro_2011_05_31 includes SO2

measurements from the Grímsvötn eruption, while the file Nabro_2011_06_05 includes measurements from both Grímsvötn and Puyehue. Including this twice in the dataset is a little unnecessary and means the user has to download more data than is needed for this eruption. It is possible to see plumes from different eruptions in many of the datafiles.

Technical Comments/Suggestions

MANUSCRIPT

Throughout – Some of the volcano names have accents (e.g. Grímsvötn, Eyjafjalla-jökull, Puyehue-Cordón Caulle)

Line 16. 'Forecast' should be forecasting or forecasts

Line 17. 'Single events' would be more precise as 'single eruptive events'

Line 17. '. . . but not any archive is available' need rewording. Perhaps: '. . . no such archive is available'.

Line 18. 'from three different instruments' would be clearer as 'from three different satellite instruments'

Line 19. 'the atmospheric parameters vertical profiles from. . .' This line is a little confusing. Perhaps rephrasing as something like: 'vertical atmospheric profiles obtained from. . .'

Line 21. 'We additionally' would read better as 'Additionally we'

Line 22. 'The dataset consists of 223 days monitored with SO2 clouds' This line does not read very well – consider rephrasing it.

Line 38-39. What is meant by 'consequent cloud'? – are you referring to the volcanic cloud or ice/water clouds (e.g. indirect climate effects)

Line 40. 'SO2 injections in the stratosphere' may read better as 'SO2 injections into

the stratosphere'

Line 42. 'hence transported' may read better as 'hence be transported'

Line 46. 'has occurred per year since 1994 worldwide' might read better as 'have occurred worldwide each year since 1994'

Line 47-48. '. . . the energy of the eruption, amount, type and size of the ejected material' would read better as '. . . the energy of the eruption, and the amount, type and size of the ejected material'

Line 49-51. To improve sentence clarity move the Newhall and Self reference to the start of the sentence: 'The VEI was introduced in 1982 by Newhall and Self (1982) . . .'

Line 50. I think it is Richter scale rather than Richter's scale.

Line 50. I think it should be earthquake rather than earthquakes'

Line 54. 'VEI index' can just be VEI

Line 60. Putting 'e.g. VEI 4 events' within brackets would help the readability of the sentence

Line 71. 'and' should be used instead of 'or'

Line 72. 'although' would make more sense than 'even though'

Line 74. 'focusing on single or a few eruptions' would read better as 'focusing on a single or a few eruptions'

Line 77. Stating that 'all' platforms and algorithms were studied in this volume seems quite strong. Perhaps: 'a large number' would be better

Lines 77 and 81. Starting the sentence with 'Sarychev Peak 2009' and 'Grimsvotn 2011' does not read very well. It might sound better as 'The Sarychev Peak eruption in 2009. . .' etc.
Line 91. '. . . and updated in the course of the years.' This line does not read very well – consider rephrasing.

Line 110. It should read '... and humidity from GNSS RO profiles'

Line 111-112. This sentence would benefit from being rewritten to improve the clarity. Maybe something like: 'This information is provided for eruptions, after 2006, classified by the GVP as VEI 4 or larger and with an SO2 mass loading of greater than 0.05 Tg. At the time of archive preparation, no eruptions after 2016 had yet been classified as VEI 4 or greater.'

Line 113. Rather than include '(table 1)' in this sentence, perhaps add a sentence at the end of the paragraph saying 'Further information on these eruptions can be found in table 1.'

Line 117-118. 'there is no current unique database'. This does not read very well – I would suggest rewriting the sentence

Line 119-121. It should read 'accurate knowledge of volcanic SO2 cloud concentration and altitude as well as their spatial and temporal evolution. . . of an eruption's climatic impact'

Line 122. 'retrievals' should be 'retrieval'

Section 2 – title. Maybe this should be titled 'Instrument and Retrieval Description'

Line 126. 'due to their own limitations.' It is not clear what is meant by this.

Line 130. It should read 'an ascending orbit'

Line 133. It would be good for a reference to be included for this sentence so the reader is immediately aware of which paper describes this technique.

Line 135. Again it would be good to have a reference for this statement.

Line 142. It should read 'an ascending orbit'

Line 144. V3 has not been defined. SO2 is also not formatted correctly.

Line 151. Slight inconsistency - On board has a space here but elsewhere it is written onboard.

Line 153. Slight inconsistency - Here the pixel size is listed as 40x80 km. For AIRS it was written as 13.5 x 13.5 km (with spaces).

Line 165. 1,67 should be 1.67

Line 175-176. It may read better as 'In this archive we use the RO bending . . .'

Line 192. It should read 'the number of days' rather than 'amount of days'

Line 232-233: 'Four of those types are of interest for this archive: type 2, 6, 9 and 10 respectively corresponding to dust, elevated smoke, volcanic ash and sulfate/other.' Include a colon between archive and type.

Line 234-235. There should be a space between 20.2/30.1 and km

Line 240. I think a colon would be better than a comma between provided and latitude

Line 264. Should read 'Where $\alpha$ is the bending angle anomaly...'

Line 277. Should read 'consists of' rather than 'consists in'

Line 285. I think this should be 'Thus we' rather than 'We thus'

Line 308. Having a list (in brackets) of parameters that affect the uncertainty, followed by a line about altitude affecting the uncertainty does not read so well. Maybe combine all the factors that affect the uncertainty into one line.

Line 315. Should read 'volcanic cloud' detection rather than 'volcanic clouds'

Line 317. It should read 'in charge of processing them'

Line 317. Is it not 10 VEI 4 and 1 VEI 5 eruptions? The Puyehue eruption in 2011 is listed on the GVP as VEI 5 (https://volcano.si.edu/volcano.cfm?vn=357150 ; under

eruptive history). Also here you state the period you a looking at is 2008 to 2016 when previously you've said you were looking at eruptions from 2006-2016.

Line 318. 'With a total of' rather than 'for a total of'

Line 320-321. 'Several parameters are measured using different instruments, such as SO2 VCD and cloud top altitude, to allow cross correlation between the different retrieval algorithms.' Do you mean to say 'Several parameters are included within the dataset... to allow cross correlation between the different algorithms' ?

Line 335. It should read 'compared the date, time ...'

Line 336. 'We have additionally' would read better as 'Additionally we have'

Line 360. Not just detection but also the retrieval of VCDs

Line 365. 'Up to date' would read better as 'At present'

Line 370. 'and test new algorithms contributing to improving the accuracy on the estimation of fundamental volcanic clouds parameters'. This may read better as 'and test new algorithms on, thereby contributing to improving the accuracy on the estimation of fundamental volcanic clouds parameters'

Line 373. 'allowing to reconstruct...' may sound better as 'allowing the reconstruction of...'

Table 2. Maybe differentiate between AIRS and IASI spatial resolutions (13.5 by 13.5 km vs. 12 km diameter circular pixels)

Figure 2. In the caption 'upright' should be 'top right'. Also, this caption reads a little strangely. I would suggest: 'Example of data use and data collocation. (a) Kasatochi cloud on 9th August 2008; (b) Sarychev peak cloud ...' At present there is no (b). Additionally, no full stop is required in line 610.

SUPPLEMENT AND DATA

P1. VC – is undefined in the supplementary material.

P1. DATE_IASI – The use of the word 'because' in this description doesn't make sense.

P2, P9. In the GOME_lon variable- dimensions include GOME_late rather than GOME_lat
* * *

---

## Referee Comment (RC2) · Anonymous Referee #2 · 29 Aug 2020

I encourage the authors to re-submit. The new data archive promises to be of great value to studies of volcanic plumes. The current version of the manuscript does not demonstrate this potential, and a revised version will encourage more volcanologists to download and use the archive.

Please also note the supplement to this comment:
https://essd.copernicus.org/preprints/essd-2020-109/essd-2020-109-RC2-supplement.pdf

2020.

**Supplement:**

Reviewer Comments

Manuscript ESSD-2020-109, Taurnigand et al., A multi-sensor satellite-based archive of the largest SO2 volcanic eruptions since 2006

The authors describe a new data archive that combines synoptic maps of $SO_2$ plumes, derived from UV and TIR radiance measurements, with estimates of plume heights and thickness derived from lidar profiles and radio occultation (RO). This archive will prove to be of great value to studies of gas emissions for the selected eruptions, as accurate knowledge of the height and thickness of plumes is critical to the estimation of gas concentrations. The authors have done a great service by collocating the RO products with the UV, TIR, and lidar data products, thus facilitating the incorporation of RO products into future analyses of the combined data record.

Unfortunately, this manuscript falls short as a description of the new data archive and showcase for the potential applications of the combined data records. With few exceptions, the descriptions of the data products are too sparse to be of much use to anyone but the most experienced remote sensing specialists. The exceptions to the sparse descriptions are focused on the RO products and, to a lesser extent, lidar profiles. Despite the attention paid to the RO data processing, the authors fail to discuss the basic relationships between the RO signals and volcanic plume phenomena. For example, what are the bending angles and bending angle anomalies (Sections 3.5, 4.1.2, and 4.1.3) and what do these anomalies tell us about volcanic plumes? Similarly, what is the relation between the RO refractivity and plume heights?

The processing of the CALIOP (lidar) data is discussed in the context of the RO data (Section 4.2.1), and the shortcomings outlined above affect the CALIOP discussion as well. The procedure for removing "unnecessary" information from the CALIOP images is confusing. How do the authors determine the "zones of interest" upon which to focus their processing? The practice of discarding CALIOP data corresponding to altitudes below 10 km seems short-sighted.  If the RO data products are noisy for such altitudes, then shouldn't the lidar data for low altitude (< 10 km) clouds be all the more important?  Since the authors never explain the significance of the RO bending anomaly, relative to volcanic plume heights, it is difficult to appreciate why the low-altitude lidar data should be discarded. Similarly, the cloud aspect ratio is not explained, and the threshold value of 0.09 is not justified. What is the significance of a low aspect ratio? What are typical aspect ratios of volcanic plumes vs. meteorological clouds? Finally, the authors do not include CALIOP depolarization ratios in the archive. The CALIOP Aerosol Type (included in the Archive) does not identify volcanic ash or sulfates uniquely, while the depolarization ratios document variations in the size and aspect of particles within volcanic plumes.

In an effort to justify the creation of the archive, the authors make a number of problematic statements. The statement that "… not any archive is available at the moment to be used as background for future studies" (Line 17) is not true, as the authors demonstrate by citing the existing LaMEVE (Lines 68-70), OMI (Lines 92-93), and TOMS/OMI MSVOLSO2L4 (Lines 95-96) archives. In addition, the authors acknowledge the GVN archive at many locations in the text. The new archive may be the first to include RO data, but the potential contributions of RO to volcanology (see my previous comments) have never been discussed. Consequently, the unique nature of the new archive is not obvious.

The authors claim that papers describing individual eruption events "make it difficult to compare" the data sets (Lines 74-75) is disingenuous. The authors neglect to mention that the studies cited in this paragraph (Lines 74-90) The authors present no evidence for the claim that the volcanic clouds generated by the Merapi, Tolbachik, Kelut, and Calbuco eruptions were not studied "in depth" (Lines 87 – 90) What is the definition of "in depth?" The Calbuso eruption clouds, in particular, have been studied extensively, as this eruption had an impact of the evolution of the southern ozone hole.

The authors need to include at least one example of unique contribution of the new archive to plume studies. Figure 2 shows the locations of data points, but not the unique contributions to the study of plumes enabled by the new archive. For example, what are the levels of agreement between the UV- and TIR-based $SO_2$ retrievals? What are the variations in $SO_2$ retrievals relative to atmospheric conditions (principally temperature and humidity) and plume altitude? What are the levels of agreement between the IASI, CALIOP, and RO-based plume altitude estimates? Which altitude estimates have the highest level of confidence?

Examples of the potential contribution of the new archive to plume studies would help with the troublesome justifications for the new archive. The contributions would become readily apparent, eliminating the need for the current unsupported statements.

---

## Author Comment (AC1) · 7 Oct 2020

We would like to thank the anonymous reviewer for the insightful and constructive comments, which helped us to improve our manuscript. We appreciate the valuable comments and try to address the issues raised as best as possible.

Specific Comments

**MANUSCRIPT**

Reviewer: Line 17. I feel it would be beneficial to elaborate here on there being no other archive which compiles the results from multiple satellite instruments and eruptions to really emphasise the strength of this dataset.

**See after**

Reply: We now emphasized it in different sections of the paper.

Reviewer: Line 20. 'We've archived and collocated ... the vertical backscatter from CALIOP ...' – I think rather than 'vertical backscatter' this sentence should indicate that you have included the CALIOP height and aerosol type.

Reply: Corrected.

Reviewer: Line 25. Here you state that 'the cross-comparison of the datasets shows the high consistency of the parameters estimated with different sensors and algorithms'. This feels like quite a strong statement. In section 5 you compare the heights obtained with RO, CALIOP and IASI. You note that for a number of eruptions there is a good agreement between RO and CALIOP but that this is not the case for Calbuco. Table 4 shows that a number of the average differences between IASI/CALIOP and IASI/RO are greater than 3 km. Additionally, you have done no quantitative comparison of the partial column densities from AIRS, IASI and GOME-2. Some rewording would improve this statement.

Reply: The current paper introduces a new data archive that combines several satellite data-sets for recent eruptions and, for the first time, includes radio occultation data. Some limited inter-comparisons of the data are already published in the literature (Brenot et al., 2014; Carn et al., 2015; Theys et al., 2013), so here we concentrate on describing the archive. Future papers are planned to inter-compare different estimations of partial column densities and cloud top heights. We reworded the sentence to make it clearer and it now reads:

**"the cross-comparison of the datasets shows different consistency of the parameters estimated with different sensors and algorithms according to the sensitivity and resolution of the instruments"**

- Brenot, H., Theys, N., Clarisse, L., van Geffen, J., van Gent, J., Van Roozendael, M., et al.: Support to Aviation Control Service (SACS): an online service for near-real-time satellite monitoring of volcanic plumes, in: Natural Hazards and Earth System Sciences, 14(5), 1099–1123. https://doi.org/10.5194/nhess-14-1099-2014, 2014. - Carn, S. A., K. Yang, A. J. Prata, and N. A. Krotkov (2015), Extending the long-term record of volcanic SO2 emissions with the Ozone Mapping and Profiler Suite nadir mapper, Geophys. Res. Lett., 42, doi:10.1002/2014GL062437.

- Theys, N., Campion, R., Clarisse, L., Brenot, H., van Gent, J., Dils, B., Corradini, S., Merucci, L., Coheur, P.-F., Van Roozendael, M., Hurtmans, D., Clerbaux, C., Tait, S., and Ferrucci, F.: Volcanic SO2 fluxes derived from satellite data: a survey using OMI, GOME-2, IASI and MODIS, Atmos. Chem. Phys., 13, 5945–5968, https://doi.org/10.5194/acp-13-5945-2013, 2013.

Reviewer: Line 91-102. This paragraph is a little confusing. You cite three papers/datasets: Ge et al. (2016); Carn et al. (2017); Carn et al. (2019) and it is a little difficult to tell if and how these papers/datasets are connected. Also, you suggest that Carn et al. (2017) included 'passive degassing' and 'main eruptive events' (line 94) but to the best of my knowledge this paper generates long term averaged fluxes that exclude large eruptive events. I would advise some rewording of this paragraph and perhaps some expansion on what is included in the Ge et al. (2016) and Carn et al. (2017) papers which might to help the reader better understand their content.

Reply: The paragraph has been reformulated as follows.

"Considering SO2 emissions, several datasets and inventories are available and updated over time, but generally include daily or yearly total emissions per volcano or per eruption. Ge et al. (2016) compiled an inventory for daily SO2 emissions in the time frame 2005-2012 including global volcanic eruptions but also eight persistently degassing volcanoes retrieved by the Ozone Monitoring Instrument (OMI) on board the Aura satellite. Carn et al. (2017) implemented it including OMI retrievals from 2005 to 2015 of emissions related to passive degassing. The most updated ... provided (Carn et al., 2016; Carn, 2019). The above-mentioned datasets provide important information for users mainly needing to assess the climatic impact of SO2 from volcanic sources, however, none of them allows for mapping the SO2 emissions and related altitude estimations in space and time and thus the direct testing and comparison of new models and techniques, like GNSS RO, for example. We think it is important to provide a complementary multi-satellite archive covering the largest eruptive events and their development all around the world in order to facilitate the access to such data for future studies."

Details about Ge et al. (2016) inventory are available at http://wiki.seas.harvard.edu/geos-chem/index.php/Volcanic\_SO2\_emissions

Reviewer: Section 2. In this section it would be useful to have some more information about the performance of each technique. For example, conditions in which the technique performs well or badly. And information such as the detection limits and uncertainties. This has been done for AIRS (lines 135-138) and something similar for each instrument/technique would help the reader appreciate the strengths and limitations of each tool. It would also help a user to correctly interpret the archived data- especially if they are comparing the results from different instruments. Section 4.3 does point the reader to some of the relevant literature but it would be nice to have this in section 2 and with more detail.

Reply: For IASI limitation and uncertainties please see the next reply.

For GOME limitation and uncertainties we added the following text to the paragraph 2.3:

"The volcanic emission measurement is facilitated by large SO2 columns generally at high altitudes (freetroposphere to lower stratosphere). However, for large SO2 columns (typically>50 DU) the absorption tends to saturate leading to a general underestimation and directly affecting the product accuracy. For most volcanoes, there is no ground-based equipment to measure SO2 during the eruption and the validation approach is usually a cross-comparisons with other satellite products. The O3M SAF validation report (Theys and Koukouli, 2015) shows that GOME-2 SO2 product reaches the target/optimal accuracy of 50%/30% respectively. It is important to notice that the SO2 retrievals from GOME-2 are also affected by clouds and instrumental noise especially at high solar zenith angles. These limitations have been filtered in the data used in this work, according to the criteria shown by Brenot et al. (2014)."

- Theys and Koukouli, https://cdop.aeronomie.be/ProjectDir/documents/ValidationReports/Validation\_Report\_GOME- 2 SO2 GDP4.8 Dec2015.pdf - Brenot, H., Theys, N., Clarisse, L., van Geffen, J., van Gent, J., Van Roozendael, M., et al.: Support to Aviation Control Service (SACS): an online service for near-real-time satellite monitoring of volcanic plumes, in: Natural Hazards and Earth System Sciences, 14(5), 1099–1123. https://doi.org/10.5194/nhess-14-1099-2014, 2014.

Reviewer: Line 144. You mention the IASI retrieval technique is based on a BTD with the v3 absorption band – brightness temperature difference with what?

Line 144-149. Initially it is implied that the IASI VCD retrieval is run using fixed heights. But in the archived data there is only a single value for the IASI VCD. Could you clarify if this is obtained by interpolating the results with the height from the second retrieval?

Line 148. It would be useful if there was a line here explaining how the IASI height retrieval worked.

Reply: Thank you for these comments; we have now updated the relevant paragraph 2.2 clarifying these different aspects (including some general statement on sensitivity and uncertainties):

"The Infrared Atmospheric Sounding Interferometer (IASI) is a Fourier transform instrument onboard the nearpolar sun-synchronous orbiting satellites Metop-A and Metop-B, respectively, launched in October 2006 and September 2012 with ascending equator crossing local time at 9:30. IASI covers the full globe two times per day with a swath of 2200 km and a spatial resolution of 12 km at nadir (Clerbaux et al., 2009). The SO2 retrieval is based on a brightness temperature difference between channels in and outside the SO2 v3 band (Clarisse et al., 2012) which is converted to SO2 concentration integrated along the vertical axis the Vertical Column Density (VCD) using look-up tables and operational profiles of pressure, temperature and humidity. The retrieval of VCD assumes that all SO2 is located at particular atmospheric layers (5, 7, 10, 13, 16, 19, 25 or 30 km above sea level) providing different estimations at different altitudes. It has a detection limit of around 0.5 DU at the tropopause, which increases for decreasing altitude (depending on the amount of water vapour in the atmosphere). For plumes above 500hPa (about 5.5 km) the algorithm has a theoretical uncertainty between 3-6%. A second algorithm (Clarisse et al., 2014) is applied to compute the SO2 cloud altitude with an accuracy of about 2 km for plumes below 20 km. The algorithm exploits the fact that the SO2 v3 band interferes with strong water vapour absorptions, and that these interferences, by virtue of the vertical water vapour profile, have a strong dependency with height. Combining the two datasets, a single bestestimate VCD is obtained by interpolating the VCD columns of the first algorithm at the retrieved height."

Reviewer: Section 2.4. I think in this section it is important to highlight that with CALIOP you are not measuring SO2 but ash, sulphate, smoke and/or dust. It would be good to acknowledge here some of the limitations of assuming SO2/ash are collocated.

Reply: CALIOP instrument does not allow SO2 measurements but dust, elevated smoke, volcanic ash and sulfate. However, the CALIOP classification algorithm do not include the volcanic ash type below the tropopause level (Kim et al. 2018) so it is difficult to distinguish the volcanic ash from other aerosol types in the lower troposphere. Both ash and SO2 are not necessarily collocated during an eruption, this is the reason why all the CALIOP data have also been collocated with the SO2 estimation from AIRS, IASI and GOME-2. We added a new sentence in the section 2.4 of the manuscript stating:

"The CALIOP does not allow SO2 measurements or estimation (it provides estimations of dust, elevated smoke, volcanic ash and sulfate) and the CALIOP classification algorithm do not include the volcanic ash type below the tropopause level (Kim et al. 2018) making difficult to distinguish the volcanic ash from other aerosol types in the lower troposphere., For these reasons, the selected CALIOP backscatter is collocated with the SO2 estimation from AIRS, IASI and GOME-2 and this combination provides a complete information on the content and vertical structure of the cloud."

- Kim, M. H., Omar, A. H., Tackett, J. L., Vaughan, M. A., Winker, D. M., Trepte, C. R., ... & Kar, J. (2018). The CALIPSO version 4 automated aerosol classification and lidar ratio selection algorithm. Atmospheric measurement techniques, 11(11), 6107.

Reviewer: Section 3. This section details all the variables contained in the files. I think it would be really beneficial to a user to have these listed in a table (either in this paper or in the supplementary information).

I found I referred to the supplementary information (print out of all the variables) a lot while trying to load and plot the data. A table summarising the variable names, meaning, dimensions, type and units would be even more useful as a quick reference guide.

Reply: We added a table with all the info.

[revised manuscript text omitted]

|                                                                                                                                                                                                                              | altitude -0.5 to 8.2 km, 8.2 to 20.2km                                                                                                                                                                                                                                                                                                                                                                                                                                                                                                                                                                                                                                                                                                                                                                                                                                                                                                      |                                                                                                                                                                                                                                                                                                                                                                             |                                                                    |                                                                                                         |
|------------------------------------------------------------------------------------------------------------------------------------------------------------------------------------------------------------------------------|---------------------------------------------------------------------------------------------------------------------------------------------------------------------------------------------------------------------------------------------------------------------------------------------------------------------------------------------------------------------------------------------------------------------------------------------------------------------------------------------------------------------------------------------------------------------------------------------------------------------------------------------------------------------------------------------------------------------------------------------------------------------------------------------------------------------------------------------------------------------------------------------------------------------------------------------|-----------------------------------------------------------------------------------------------------------------------------------------------------------------------------------------------------------------------------------------------------------------------------------------------------------------------------------------------------------------------------|--------------------------------------------------------------------|---------------------------------------------------------------------------------------------------------|
|                                                                                                                                                                                                                              | and 20.2 to 30.1km                                                                                                                                                                                                                                                                                                                                                                                                                                                                                                                                                                                                                                                                                                                                                                                                                                                                                                                          |                                                                                                                                                                                                                                                                                                                                                                             |                                                                    |                                                                                                         |
|                                                                                                                                                                                                                              | Only volcano files                                                                                                                                                                                                                                                                                                                                                                                                                                                                                                                                                                                                                                                                                                                                                                                                                                                                                                                          |                                                                                                                                                                                                                                                                                                                                                                             |                                                                    |                                                                                                         |
| DO 1                                                                                                                                                                                                                         | Latitude data, each row corresponds to                                                                                                                                                                                                                                                                                                                                                                                                                                                                                                                                                                                                                                                                                                                                                                                                                                                                                                      |                                                                                                                                                                                                                                                                                                                                                                             |                                                                    | 1 4                                                                                                     |
| KO_lat                                                                                                                                                                                                                       | one profile point and each column to a                                                                                                                                                                                                                                                                                                                                                                                                                                                                                                                                                                                                                                                                                                                                                                                                                                                                                                      | RO_lat, RO_profile                                                                                                                                                                                                                                                                                                                                                          | double                                                             | degrees north                                                                                           |
|                                                                                                                                                                                                                              | ro profile.                                                                                                                                                                                                                                                                                                                                                                                                                                                                                                                                                                                                                                                                                                                                                                                                                                                                                                                                 |                                                                                                                                                                                                                                                                                                                                                                             |                                                                    |                                                                                                         |
| RO lon                                                                                                                                                                                                                       | one profile point and each column to a                                                                                                                                                                                                                                                                                                                                                                                                                                                                                                                                                                                                                                                                                                                                                                                                                                                                                                      | RO lat RO profile                                                                                                                                                                                                                                                                                                                                                           | double                                                             | degrees east                                                                                            |
| KO_IOII                                                                                                                                                                                                                      | ro profile                                                                                                                                                                                                                                                                                                                                                                                                                                                                                                                                                                                                                                                                                                                                                                                                                                                                                                                                  | KO_lat, KO_profile                                                                                                                                                                                                                                                                                                                                                          | uouoie                                                             | uegrees east                                                                                            |
|                                                                                                                                                                                                                              | Date and time data each row                                                                                                                                                                                                                                                                                                                                                                                                                                                                                                                                                                                                                                                                                                                                                                                                                                                                                                                 |                                                                                                                                                                                                                                                                                                                                                                             |                                                                    | seconds since                                                                                           |
| RO date                                                                                                                                                                                                                      | corresponds to one profile point and                                                                                                                                                                                                                                                                                                                                                                                                                                                                                                                                                                                                                                                                                                                                                                                                                                                                                                        | RO lat RO profile                                                                                                                                                                                                                                                                                                                                                           | int                                                                | 1970-01-01                                                                                              |
|                                                                                                                                                                                                                              | each column to a ro profile.                                                                                                                                                                                                                                                                                                                                                                                                                                                                                                                                                                                                                                                                                                                                                                                                                                                                                                                | rto_nut, rto_pronite                                                                                                                                                                                                                                                                                                                                                        |                                                                    | 00:00:0.0                                                                                               |
|                                                                                                                                                                                                                              | Bending angle data, each row                                                                                                                                                                                                                                                                                                                                                                                                                                                                                                                                                                                                                                                                                                                                                                                                                                                                                                                |                                                                                                                                                                                                                                                                                                                                                                             |                                                                    |                                                                                                         |
| RO bending angle                                                                                                                                                                                                             | corresponds to one profile point and                                                                                                                                                                                                                                                                                                                                                                                                                                                                                                                                                                                                                                                                                                                                                                                                                                                                                                        | RO lat, RO profile                                                                                                                                                                                                                                                                                                                                                          | double                                                             | rad                                                                                                     |
|                                                                                                                                                                                                                              | each column to a ro profile.                                                                                                                                                                                                                                                                                                                                                                                                                                                                                                                                                                                                                                                                                                                                                                                                                                                                                                                |                                                                                                                                                                                                                                                                                                                                                                             |                                                                    |                                                                                                         |
|                                                                                                                                                                                                                              | Bending angle anomaly data, each row                                                                                                                                                                                                                                                                                                                                                                                                                                                                                                                                                                                                                                                                                                                                                                                                                                                                                                        |                                                                                                                                                                                                                                                                                                                                                                             |                                                                    |                                                                                                         |
| RO_anomaly_bending_angle                                                                                                                                                                                                     | corresponds to one profile point and                                                                                                                                                                                                                                                                                                                                                                                                                                                                                                                                                                                                                                                                                                                                                                                                                                                                                                        | RO_lat, RO_profile                                                                                                                                                                                                                                                                                                                                                          | double                                                             | percent                                                                                                 |
|                                                                                                                                                                                                                              | each column to a ro profile.                                                                                                                                                                                                                                                                                                                                                                                                                                                                                                                                                                                                                                                                                                                                                                                                                                                                                                                |                                                                                                                                                                                                                                                                                                                                                                             |                                                                    |                                                                                                         |
|                                                                                                                                                                                                                              | Temperature data, each row corresponds                                                                                                                                                                                                                                                                                                                                                                                                                                                                                                                                                                                                                                                                                                                                                                                                                                                                                                      |                                                                                                                                                                                                                                                                                                                                                                             |                                                                    |                                                                                                         |
| RO_temperature                                                                                                                                                                                                               | to one profile point and each column to                                                                                                                                                                                                                                                                                                                                                                                                                                                                                                                                                                                                                                                                                                                                                                                                                                                                                                     | RO_lat, RO_profile                                                                                                                                                                                                                                                                                                                                                          | double                                                             | K                                                                                                       |
|                                                                                                                                                                                                                              | a ro profile.                                                                                                                                                                                                                                                                                                                                                                                                                                                                                                                                                                                                                                                                                                                                                                                                                                                                                                                               |                                                                                                                                                                                                                                                                                                                                                                             |                                                                    |                                                                                                         |
|                                                                                                                                                                                                                              | Pressure data, each row corresponds to                                                                                                                                                                                                                                                                                                                                                                                                                                                                                                                                                                                                                                                                                                                                                                                                                                                                                                      |                                                                                                                                                                                                                                                                                                                                                                             |                                                                    |                                                                                                         |
| RO_pressure                                                                                                                                                                                                                  | one profile point and each column to a                                                                                                                                                                                                                                                                                                                                                                                                                                                                                                                                                                                                                                                                                                                                                                                                                                                                                                      | RO_lat, RO_profile                                                                                                                                                                                                                                                                                                                                                          | double                                                             | Pa                                                                                                      |
|                                                                                                                                                                                                                              | ro profile.                                                                                                                                                                                                                                                                                                                                                                                                                                                                                                                                                                                                                                                                                                                                                                                                                                                                                                                                 |                                                                                                                                                                                                                                                                                                                                                                             | ļ                                                                  |                                                                                                         |
|                                                                                                                                                                                                                              | Refractivity data, each row corresponds                                                                                                                                                                                                                                                                                                                                                                                                                                                                                                                                                                                                                                                                                                                                                                                                                                                                                                     |                                                                                                                                                                                                                                                                                                                                                                             | 1 11                                                               | 1                                                                                                       |
| RO_refractivity                                                                                                                                                                                                              | to one profile point and each column to                                                                                                                                                                                                                                                                                                                                                                                                                                                                                                                                                                                                                                                                                                                                                                                                                                                                                                     | RO_lat, RO_profile                                                                                                                                                                                                                                                                                                                                                          | double                                                             | 1                                                                                                       |
|                                                                                                                                                                                                                              | a ro profile.                                                                                                                                                                                                                                                                                                                                                                                                                                                                                                                                                                                                                                                                                                                                                                                                                                                                                                                               |                                                                                                                                                                                                                                                                                                                                                                             |                                                                    |                                                                                                         |
| PO specific humidity                                                                                                                                                                                                         | specific number data, each fow                                                                                                                                                                                                                                                                                                                                                                                                                                                                                                                                                                                                                                                                                                                                                                                                                                                                                                              | PO lat PO profile                                                                                                                                                                                                                                                                                                                                                           | double                                                             | ka ka 1                                                                                                 |
| KO_specific_numberry                                                                                                                                                                                                         | each column to a ro profile                                                                                                                                                                                                                                                                                                                                                                                                                                                                                                                                                                                                                                                                                                                                                                                                                                                                                                                 | KO_lat, KO_profile                                                                                                                                                                                                                                                                                                                                                          | uouoie                                                             | Kg.Kg-1                                                                                                 |
|                                                                                                                                                                                                                              | Cloud top altitude data each column                                                                                                                                                                                                                                                                                                                                                                                                                                                                                                                                                                                                                                                                                                                                                                                                                                                                                                         |                                                                                                                                                                                                                                                                                                                                                                             |                                                                    |                                                                                                         |
| RO_heightVC                                                                                                                                                                                                                  | corresponds to a ro profile                                                                                                                                                                                                                                                                                                                                                                                                                                                                                                                                                                                                                                                                                                                                                                                                                                                                                                                 | 1, RO_profile                                                                                                                                                                                                                                                                                                                                                               | double                                                             | m                                                                                                       |
|                                                                                                                                                                                                                              | Only daily files                                                                                                                                                                                                                                                                                                                                                                                                                                                                                                                                                                                                                                                                                                                                                                                                                                                                                                                            |                                                                                                                                                                                                                                                                                                                                                                             | 4                                                                  |                                                                                                         |
|                                                                                                                                                                                                                              | Latitude data, each row corresponds to                                                                                                                                                                                                                                                                                                                                                                                                                                                                                                                                                                                                                                                                                                                                                                                                                                                                                                      |                                                                                                                                                                                                                                                                                                                                                                             |                                                                    |                                                                                                         |
| RO AIRS lat                                                                                                                                                                                                                  | one profile point and each column to a                                                                                                                                                                                                                                                                                                                                                                                                                                                                                                                                                                                                                                                                                                                                                                                                                                                                                                      | RO_AIRS_lat,                                                                                                                                                                                                                                                                                                                                                                | double                                                             | degrees north                                                                                           |
|                                                                                                                                                                                                                              | ro profile.                                                                                                                                                                                                                                                                                                                                                                                                                                                                                                                                                                                                                                                                                                                                                                                                                                                                                                                                 | RO_AIRS_profile                                                                                                                                                                                                                                                                                                                                                             |                                                                    | 0                                                                                                       |
|                                                                                                                                                                                                                              | Longitude data, each row corresponds to                                                                                                                                                                                                                                                                                                                                                                                                                                                                                                                                                                                                                                                                                                                                                                                                                                                                                                     | DO AIDS lat                                                                                                                                                                                                                                                                                                                                                                 |                                                                    |                                                                                                         |
| RO_AIRS_lon                                                                                                                                                                                                                  | one profile point and each column to a                                                                                                                                                                                                                                                                                                                                                                                                                                                                                                                                                                                                                                                                                                                                                                                                                                                                                                      | RO_AIRS_rat,                                                                                                                                                                                                                                                                                                                                                                | double                                                             | degrees east                                                                                            |
|                                                                                                                                                                                                                              | ro profile.                                                                                                                                                                                                                                                                                                                                                                                                                                                                                                                                                                                                                                                                                                                                                                                                                                                                                                                                 | KO_AIKS_pionic                                                                                                                                                                                                                                                                                                                                                              | ļ                                                                  |                                                                                                         |
|                                                                                                                                                                                                                              | Date and time data, each row                                                                                                                                                                                                                                                                                                                                                                                                                                                                                                                                                                                                                                                                                                                                                                                                                                                                                                                | RO AIRS lat                                                                                                                                                                                                                                                                                                                                                                 |                                                                    | seconds since                                                                                           |
| RO_AIRS_date                                                                                                                                                                                                                 | corresponds to one profile point and                                                                                                                                                                                                                                                                                                                                                                                                                                                                                                                                                                                                                                                                                                                                                                                                                                                                                                        | RO AIRS profile                                                                                                                                                                                                                                                                                                                                                             | int                                                                | 1070 01 01                                                                                              |
|                                                                                                                                                                                                                              | each column to a ro profile.                                                                                                                                                                                                                                                                                                                                                                                                                                                                                                                                                                                                                                                                                                                                                                                                                                                                                                                |                                                                                                                                                                                                                                                                                                                                                                             | m                                                                  | 19/0-01-01                                                                                              |
|                                                                                                                                                                                                                              |                                                                                                                                                                                                                                                                                                                                                                                                                                                                                                                                                                                                                                                                                                                                                                                                                                                                                                                                             | F                                                                                                                                                                                                                                                                                                                                                                           | IIIt                                                               | 1970-01-01
00:00:0.0                                                                                 |
|                                                                                                                                                                                                                              | Bending angle data, each row                                                                                                                                                                                                                                                                                                                                                                                                                                                                                                                                                                                                                                                                                                                                                                                                                                                                                                                | RO AIRS lat,                                                                                                                                                                                                                                                                                                                                                                | 1.11                                                               | 00:00:0.0                                                                                               |
| RO_AIRS_bending_angle                                                                                                                                                                                                        | Bending angle data, each row
corresponds to one profile point and                                                                                                                                                                                                                                                                                                                                                                                                                                                                                                                                                                                                                                                                                                                                                                                                                                                                        | RO_AIRS_lat,
RO_AIRS_profile                                                                                                                                                                                                                                                                                                                                             | double                                                             | rad                                                                                                     |
| RO_AIRS_bending_angle                                                                                                                                                                                                        | Bending angle data, each row corresponds to one profile point and each column to a ro profile.                                                                                                                                                                                                                                                                                                                                                                                                                                                                                                                                                                                                                                                                                                                                                                                                                                              | RO_AIRS_lat,
RO_AIRS_profile                                                                                                                                                                                                                                                                                                                                             | double                                                             | rad                                                                                                     |
| RO_AIRS_bending_angle                                                                                                                                                                                                        | Bending angle data, each row
corresponds to one profile point and
each column to a ro profile.
Bending angle anomaly data, each row                                                                                                                                                                                                                                                                                                                                                                                                                                                                                                                                                                                                                                                                                                                                                                                                | RO_AIRS_lat,
RO_AIRS_profile
RO_AIRS_lat,                                                                                                                                                                                                                                                                                                                             | double                                                             | 1970-01-01
00:00:0.0
rad                                                                          |
| RO_AIRS_bending_angle
RO_AIRS_anomaly_bending_angle                                                                                                                                                                       | Bending angle data, each row
corresponds to one profile point and
each column to a ro profile.
Bending angle anomaly data, each row
corresponds to one profile point and
each column to a ro profile                                                                                                                                                                                                                                                                                                                                                                                                                                                                                                                                                                                                                                                                                                                         | RO_AIRS_lat,
RO_AIRS_profile
RO_AIRS_lat,
RO_AIRS_profile                                                                                                                                                                                                                                                                                                          | double                                                             | rad                                                                                                     |
| RO_AIRS_bending_angle
RO_AIRS_anomaly_bending_angle                                                                                                                                                                       | Bending angle data, each row
corresponds to one profile point and
each column to a ro profile.
Bending angle anomaly data, each row
corresponds to one profile point and
each column to a ro profile.                                                                                                                                                                                                                                                                                                                                                                                                                                                                                                                                                                                                                                                                                                                        | RO_AIRS_lat,
RO_AIRS_profile
RO_AIRS_lat,
RO_AIRS_profile                                                                                                                                                                                                                                                                                                          | double
double                                                   | rad                                                                                                     |
| RO_AIRS_bending_angle
RO_AIRS_anomaly_bending_angle                                                                                                                                                                       | Bending angle data, each row
corresponds to one profile point and
each column to a ro profile.
Bending angle anomaly data, each row
corresponds to one profile point and
each column to a ro profile.
Temperature data, each row corresponds
to one profile point and each column to                                                                                                                                                                                                                                                                                                                                                                                                                                                                                                                                                                                                                                   | RO_AIRS_lat,
RO_AIRS_profile
RO_AIRS_lat,
RO_AIRS_profile
RO_AIRS_lat,                                                                                                                                                                                                                                                                                          | double
double                                                   | rad
percent                                                                                          |
| RO_AIRS_bending_angle
RO_AIRS_anomaly_bending_angle
RO_AIRS_temperature                                                                                                                                                | Bendingangledata,eachrowcorrespondsto one profile point andeach column to a ro profile.Bending angle anomaly data, each rowcorrespondsto one profile point andeach column to a ro profile.Temperature data, each row correspondsto one profile point and each column toa ro profile.                                                                                                                                                                                                                                                                                                                                                                                                                                                                                                                                                                                                                                                        | RO_AIRS_lat,
RO_AIRS_profile
RO_AIRS_lat,
RO_AIRS_profile
RO_AIRS_lat,
RO_AIRS_lat,
RO_AIRS_profile                                                                                                                                                                                                                                                       | double
double
double                                         | rad
percent
K                                                                                     |
| RO_AIRS_bending_angle RO_AIRS_anomaly_bending_angle RO_AIRS_temperature                                                                                                                                                      | Bendingangledata,eachrowcorrespondsto one profile point andeach column to a ro profile.Bendingangle anomaly data, each rowcorrespondsto one profile point andeach column to a ro profile.Temperature data, each row correspondsto one profile point and each column toa ro profile.Pressure data, each row corresponds to                                                                                                                                                                                                                                                                                                                                                                                                                                                                                                                                                                                                                   | RO_AIRS_lat,
RO_AIRS_profile
RO_AIRS_lat,
RO_AIRS_profile
RO_AIRS_lat,
RO_AIRS_lat,
RO_AIRS_lat,                                                                                                                                                                                                                                                          | double
double
double                                         | rad
percent
K                                                                                     |
| RO_AIRS_bending_angle
RO_AIRS_anomaly_bending_angle
RO_AIRS_temperature
RO_AIRS_pressure                                                                                                                            | Bendingangledata,eachrowcorrespondsto one profile point andeach column to a ro profile.Bendingangle anomaly data, each rowcorrespondsto one profile point andeach column to a ro profile.Temperature data, each row correspondsto one profile point and each column toa ro profile.Pressure data, each row corresponds toone profile.Pressure data, each row corresponds toone profile.                                                                                                                                                                                                                                                                                                                                                                                                                                                                                                                                                     | RO_AIRS_lat,
RO_AIRS_profile
RO_AIRS_lat,
RO_AIRS_profile
RO_AIRS_lat,
RO_AIRS_profile
RO_AIRS_lat,
RO_AIRS_lat,                                                                                                                                                                                                                                       | double
double
double
double                               | rad
percent
K                                                                                     |
| RO_AIRS_bending_angle         RO_AIRS_anomaly_bending_angle         RO_AIRS_temperature         RO_AIRS_pressure                                                                                                             | Bendingangledata,eachrowcorrespondsto one profile point andeach column to a ro profile.Bendingangle anomaly data, each rowcorrespondsto one profile point andeach column to a ro profile.Temperature data, each row correspondsto one profile point and each column toa ro profile.Pressure data, each row corresponds toone profile.Pressure data, each row corresponds toone profile.Profile.Profile.                                                                                                                                                                                                                                                                                                                                                                                                                                                                                                                                     | RO_AIRS_lat,
RO_AIRS_profileRO_AIRS_lat,
RO_AIRS_profileRO_AIRS_lat,
RO_AIRS_lat,
RO_AIRS_profileRO_AIRS_lat,
RO_AIRS_profile                                                                                                                                                                                                                                | double
double
double
double                               | rad
percent
K
Pa                                                                               |
| RO_AIRS_bending_angle         RO_AIRS_anomaly_bending_angle         RO_AIRS_temperature         RO_AIRS_pressure                                                                                                             | Bendingangledata,eachrowcorrespondsto one profile point andeach column to a ro profile.Bendingangle anomaly data, each rowcorrespondsto one profile point andeach column to a ro profile.Temperature data, each row correspondsto one profile point and each column toa ro profile.Pressure data, each row corresponds toone profile point and each column toa ro profile.Pressure data, each row corresponds toone profile point and each column to aro profile.Refractivity data, each row corresponds                                                                                                                                                                                                                                                                                                                                                                                                                                    | RO_AIRS_lat,
RO_AIRS_profileRO_AIRS_lat,
RO_AIRS_profileRO_AIRS_lat,
RO_AIRS_lat,
RO_AIRS_lat,
RO_AIRS_profileRO_AIRS_lat,
RO_AIRS_lat,
RO_AIRS_profile                                                                                                                                                                                                | double
double
double                                         | rad
percent
K
Pa                                                                               |
| RO_AIRS_bending_angle         RO_AIRS_anomaly_bending_angle         RO_AIRS_temperature         RO_AIRS_pressure         RO_AIRS_refractivity                                                                                | Bendingangledata,eachrowcorrespondsto one profile point andeach column to a ro profile.Bendingangle anomaly data, each rowcorrespondsto one profile point andeach column to a ro profile.Temperature data, each row correspondsto one profile point and each column toa ro profile.Pressure data, each row corresponds toone profile point and each column to aro profile.Refractivity data, each row correspondsto one profile point and each column to                                                                                                                                                                                                                                                                                                                                                                                                                                                                                    | RO_AIRS_lat,
RO_AIRS_profileRO_AIRS_lat,
RO_AIRS_profileRO_AIRS_lat,
RO_AIRS_profileRO_AIRS_lat,
RO_AIRS_lat,
RO_AIRS_profileRO_AIRS_lat,
RO_AIRS_profileRO_AIRS_profile                                                                                                                                                                                  | double
double
double
double
double                     | 1970-01-01       00:00:0.0       rad       percent       K       Pa       1                             |
| RO_AIRS_bending_angle         RO_AIRS_anomaly_bending_angle         RO_AIRS_temperature         RO_AIRS_pressure         RO_AIRS_refractivity                                                                                | Bendingangledata,eachrowcorrespondsto one profile point andeach column to a ro profile.Bendingangle anomaly data, each rowcorrespondsto one profile point andeach column to a ro profile.Temperature data, each row correspondsto one profile point and each column toa ro profile.Pressure data, each row corresponds toone profile point and each column to aro profile.Refractivity data, each row correspondsto one profile point and each column to aro profile.Refractivity data, each row correspondsto one profile point and each column toa ro profile.                                                                                                                                                                                                                                                                                                                                                                            | RO_AIRS_lat,
RO_AIRS_profileRO_AIRS_lat,
RO_AIRS_profileRO_AIRS_lat,
RO_AIRS_profileRO_AIRS_lat,
RO_AIRS_profileRO_AIRS_lat,
RO_AIRS_profileRO_AIRS_lat,
RO_AIRS_profileRO_AIRS_profile                                                                                                                                                                   | double
double
double
double                               | rad
percent
K
Pa
1                                                                          |
| RO_AIRS_bending_angle         RO_AIRS_anomaly_bending_angle         RO_AIRS_temperature         RO_AIRS_pressure         RO_AIRS_refractivity                                                                                | Bendingangledata,eachrowcorrespondsto one profile point andeach column to a ro profile.Bending angle anomaly data, each rowcorrespondsto one profile point andeach column to a ro profile.Temperature data, each row correspondsto one profile point and each column toa ro profile.Pressure data, each row corresponds toone profile point and each column to aro profile.Refractivity data, each row correspondsto one profile point and each column to aro profile.Refractivity data, each row correspondsto one profile point and each column toa ro profile.Specific humidity data, each row                                                                                                                                                                                                                                                                                                                                           | RO_AIRS_lat,
RO_AIRS_profileRO_AIRS_lat,
RO_AIRS_profileRO_AIRS_lat,
RO_AIRS_profileRO_AIRS_lat,
RO_AIRS_profileRO_AIRS_lat,
RO_AIRS_profileRO_AIRS_lat,
RO_AIRS_profileRO_AIRS_lat,
RO_AIRS_profileRO_AIRS_lat,
RO_AIRS_profileRO_AIRS_lat,
RO_AIRS_profileRO_AIRS_lat,
RO_AIRS_profile                                                      | double
double
double
double                               | rad
percent
K
Pa
1                                                                          |
| RO_AIRS_bending_angle         RO_AIRS_anomaly_bending_angle         RO_AIRS_temperature         RO_AIRS_pressure         RO_AIRS_refractivity         RO_AIRS_specific_humidity                                              | Bendingangledata,eachrowcorrespondsto one profile point andeach column to a ro profile.Bending angle anomaly data, each rowcorrespondsto one profile point andeach column to a ro profile.Temperature data, each row correspondsto one profile point and each column toa ro profile.Pressure data, each row corresponds toone profile point and each column to aro profile.Refractivity data, each row correspondsto one profile point and each column to aro profile.Refractivity data, each row correspondsto one profile point and each column toa ro profile.Specific humidity data, each rowcorresponds to one profile point and                                                                                                                                                                                                                                                                                                       | RO_AIRS_lat,
RO_AIRS_profileRO_AIRS_lat,
RO_AIRS_profileRO_AIRS_lat,
RO_AIRS_profileRO_AIRS_lat,
RO_AIRS_profileRO_AIRS_lat,
RO_AIRS_profileRO_AIRS_lat,
RO_AIRS_profileRO_AIRS_profileRO_AIRS_profileRO_AIRS_profileRO_AIRS_profileRO_AIRS_profileRO_AIRS_profileRO_AIRS_profileRO_AIRS_profile                                                          | double
double
double
double
double                     | 1970-01-01         00:00:0.0         rad         percent         K         Pa         1         kg.kg-1 |
| RO_AIRS_bending_angle         RO_AIRS_anomaly_bending_angle         RO_AIRS_temperature         RO_AIRS_pressure         RO_AIRS_refractivity         RO_AIRS_specific_humidity                                              | Bending
each row
corresponds to one profile point and
each column to a ro profile.Bending angle anomaly data, each row
corresponds to one profile point and
each column to a ro profile.Temperature data, each row corresponds
to one profile point and each column to
a ro profile.Pressure data, each row corresponds to
one profile point and each column to a
ro profile.Pressure data, each row corresponds to
one profile point and each column to a
ro profile.Refractivity data, each row corresponds
to one profile point and each column to
a ro profile.Refractivity data, each row corresponds
to one profile point and each column to
a ro profile.Specific humidity data, each row
corresponds to one profile point and
each column to a ro profile.                                                                                                                       | RO_AIRS_lat,
RO_AIRS_lat,
RO_AIRS_profile
RO_AIRS_profile
RO_AIRS_lat,
RO_AIRS_lat,
RO_AIRS_lat,
RO_AIRS_profile
RO_AIRS_lat,
RO_AIRS_profile
RO_AIRS_profile                                                                                                                                                                                 | double
double
double
double
double                     | 1970-01-01         00:00:0.0         rad         percent         K         Pa         1         kg.kg-1 |
| RO_AIRS_bending_angle         RO_AIRS_anomaly_bending_angle         RO_AIRS_temperature         RO_AIRS_pressure         RO_AIRS_refractivity         RO_AIRS_specific_humidity         RO_AIRS_heightVC                     | Bendingangledata,eachrowcorrespondsto one profile point andeach column to a ro profile.Bending angle anomaly data, each rowcorrespondsto one profile point andeach column to a ro profile.Temperature data, each row correspondsto one profile point and each column toa ro profile.Pressure data, each row corresponds toone profile point and each column to aro profile.Refractivity data, each row correspondsto one profile point and each column to aro profile.Refractivity data, each row correspondsto one profile point and each column toa ro profile.Specific humidity data, each rowcorresponds to one profile point andeach column to a ro profile.Cloud top altitude data, each column                                                                                                                                                                                                                                       | RO_AIRS_lat,
RO_AIRS_lat,
RO_AIRS_lat,
RO_AIRS_lat,
RO_AIRS_lat,
RO_AIRS_lat,
RO_AIRS_lat,
RO_AIRS_lat,
RO_AIRS_lat,
RO_AIRS_profile
RO_AIRS_profile
RO_AIRS_profile                                                                                                                                                                       | double
double
double
double
double
double           | rad
percent
K
Pa
1
kg.kg-1
m                                                          |
| RO_AIRS_bending_angle         RO_AIRS_anomaly_bending_angle         RO_AIRS_temperature         RO_AIRS_pressure         RO_AIRS_refractivity         RO_AIRS_specific_humidity         RO_AIRS_heightVC                     | Bendingangledata,eachrowcorrespondsto one profile pointandeach column to a ro profile.Bendingangle anomaly data, each rowcorrespondsto one profile pointandeach column to a ro profile.Temperature data, each row correspondsto one profile point and each column toa ro profile.Pressure data, each row corresponds toone profile point and each column to aro profile.Refractivity data, each row correspondsto one profile point and each column toa ro profile.Refractivity data, each row correspondsto one profile point and each column toa ro profile.Specific humidity data, each rowcorresponds to one profile point andeach column to a ro profile.Cloud top altitude data, each columncorresponds to a ro profile.Latitude dataLatitude datacorresponds to a ro profile.                                                                                                                                                        | RO_AIRS_lat,
RO_AIRS_profileRO_AIRS_lat,
RO_AIRS_profileRO_AIRS_lat,
RO_AIRS_profileRO_AIRS_lat,
RO_AIRS_profileRO_AIRS_lat,
RO_AIRS_profileRO_AIRS_lat,
RO_AIRS_profileRO_AIRS_profileI,
RO_AIRS_profile1,
RO_AIRS_profile                                                                                                                         | double
double
double
double
double
double           | rad
percent
K
Pa
1
kg.kg-1
m                                                          |
| RO_AIRS_bending_angle         RO_AIRS_anomaly_bending_angle         RO_AIRS_temperature         RO_AIRS_pressure         RO_AIRS_refractivity         RO_AIRS_specific_humidity         RO_AIRS_heightVC                     | Bending angle data, each row corresponds to one profile point and each column to a ro profile.         Bending angle anomaly data, each row corresponds to one profile point and each column to a ro profile.         Temperature data, each row corresponds to one profile point and each column to a ro profile.         Pressure data, each row corresponds to one profile point and each column to a ro profile.         Pressure data, each row corresponds to one profile point and each column to a ro profile.         Refractivity data, each row corresponds to one profile point and each column to a ro profile.         Specific humidity data, each row corresponds to one profile.         Specific humidity data, each row corresponds to one profile.         Cloud top altitude data, each column corresponds to a ro profile.         Latitude data, each row corresponds to one profile.                                | RO_AIRS_lat,
RO_AIRS_profileRO_AIRS_lat,
RO_AIRS_profileRO_AIRS_lat,
RO_AIRS_profileRO_AIRS_lat,
RO_AIRS_profileRO_AIRS_lat,
RO_AIRS_profileRO_AIRS_lat,
RO_AIRS_profileRO_AIRS_profileRO_AIRS_profileRO_AIRS_profileRO_AIRS_profileRO_AIRS_profileRO_AIRS_profileRO_AIRS_profileRO_AIRS_profileRO_AIRS_profileRO_AIRS_profileRO_AIRS_profileRO_IASI_lat, | double
double
double
double
double
double           | rad
percent
K
Pa
1
kg.kg-1
m                                                          |
| RO_AIRS_bending_angle         RO_AIRS_anomaly_bending_angle         RO_AIRS_temperature         RO_AIRS_pressure         RO_AIRS_refractivity         RO_AIRS_specific_humidity         RO_AIRS_heightVC         RO_IASI_lat | Bending angle data, each row corresponds to one profile point and each column to a ro profile.         Bending angle anomaly data, each row corresponds to one profile point and each column to a ro profile.         Temperature data, each row corresponds to one profile point and each column to a ro profile.         Pressure data, each row corresponds to one profile point and each column to a ro profile.         Pressure data, each row corresponds to one profile point and each column to a ro profile.         Refractivity data, each row corresponds to one profile point and each column to a ro profile.         Specific humidity data, each row corresponds to one profile.         Cloud top altitude data, each column to a ro profile.         Cloud top altitude data, each column corresponds to a ro profile.         Latitude data, each row corresponds to one profile point and each column to a ro profile. | RO_AIRS_lat,
RO_AIRS_profileRO_AIRS_lat,
RO_AIRS_profileRO_AIRS_lat,
RO_AIRS_profileRO_AIRS_lat,
RO_AIRS_profileRO_AIRS_lat,
RO_AIRS_profileRO_AIRS_lat,
RO_AIRS_profileRO_AIRS_lat,
RO_AIRS_profileRO_AIRS_profileRO_AIRS_profileRO_AIRS_profileRO_AIRS_profileRO_AIRS_profileRO_AIRS_profileRO_AIRS_profileRO_IASI_lat,
RO_IASI_profile           | double
double
double
double
double
double
double | rad
percent
K
Pa
1
kg.kg-1
m
degrees north                                         |

[revised manuscript text omitted]

Reviewer: Line 201. This is the first instance that 'granule' has been used. Please can this be clearly defined. The use of the word granule made it challenging to interpret the data structures described in sections 3.1-3.3 independently of reading in and looking at the data.

Reply: The term "granule" refers to the AIRS data, while, for IASI and GOME-2, we refer to scanning lines. AIRS collects data as it sweeps along the orbit, and the data is then sectioned into pieces called "granules". Each AIRS granule is roughly 2250 x 1650 kilometers and contains 6 minutes of data. There are nominally 240 Level 1B and 240 Level 2 granules of 6-minute duration generated each day. The orbital repeat cycle is 16 days, but orbital maintenance manoeuvres can shift granules along orbits by a small fraction of a granule. Maps showing the locations of granules are generated daily and available for download. AIRS data users use maps like these when making requests from AIRS data servers. We now explain the term granule in the manuscript with the following sentence:

"A granule is a portion of AIRS orbit containing 6 minutes (2250 km x 1650 km) of data, which is officially defined by the National Aeronautics and Space Administration (NASA)."

Section 3.4, Section 3.5. In these sections you mention collocation with between CALIOP, RO and the other instruments. I think it would be useful to know what conditions you use for the collocation here rather than in sections 4.1 and 4.2. This would help the reader immediately understand what is meant by collocation. Reply: Done.

Reviewer: Lines 340-344. Here you discuss the average differences between the cloud heights for different eruptions. Are there any reasons why the average difference is greater at Calbuco than for Eyja, Kasatochi and Grimsvotn? Also, is there a reason why the differences are greater between IASI and RO/CALIOP? I think it would be useful for a user of the dataset to understand why differences might arise between the different datasets (e.g. the time difference between the overpasses and the method used to obtain the height information).

Reply: The reasons are due to different eruption types and the different sensitivity and resolution of the measurement techniques. As reported in section 2.2 (IASI), some assumptions have been made to retrieve the cloud height allowing an estimation with an accuracy of about 2 km. Moreover, the IASI height estimations are sampled every 0.5 km. The RO cloud height estimation is based on the density variation of the atmosphere, so denser clouds (e.g. Kasatochi 2008) can be detected more likely than less dense clouds (e.g. Calbuco 2015) and with better accuracy. Most importantly, the RO and CALIOP are limb profiling techniques with high vertical resolution, while IASI is a nadir sounding technique. This does not allow IASI to provide the same vertical resolution and accuracy that we can get from RO/CALIOP.

In addition, the number of colocations between RO and CALIOP is much smaller than for RO-IASI and IASI-CALIOP, respectively. We revised the text at the end of section 5 (Data cross-comparisons) and added further explanations:

"The difference in cloud top estimations can be partly explained by the different sensitivities and vertical resolution of the different instruments. In addition, the number of colocations between RO and CALIOP is much smaller than for RO-IASI and IASI-CALIOP, respectively. The cloud top height estimation for eruptions with a large number of colocations (Calbuco, Kasatochi, Nabro and Sarychev Peak) is in general consistent within the techniques."

Reviewer: Additionally, have you considered a quantitative comparison of the VCDs retrieved with AIRS, IASI and GOME-2? What differences would you expect to see between these?

Reply: As we reported above, the current paper introduces a new data archive that combines several satellite data-sets for recent eruptions and, for the first time, includes radio occultation data. Some limited intercomparisons of the data are already published in the literature (Brenot et al., 2014; Carn et al., 2015; Theys et al., 2013), so here we concentrate on describing the archive. In Table 3 we report the SO2 mass loading for each eruption with different instruments reported in literature. We prefer to refer to published studies, instead of re-computing the mass loadings in these specific cases, to avoid confusion to the readers.

Reviewer: Table 1. You could add the eruption VEI and the eruption end date or duration to this table. Additionally, it could be helpful to add the geographic region considered for each eruption and the start/end date for the data in the archive – both of these would be valuable to the data user.

Reply: We added to the table the VEI and the archive start/end dates for each eruption. Please note that the VEI is not always appropriate for SO2-rich eruptions since it corresponds to ash-rich eruptions. Instead of

adding the geographic region in the table, we prefer to provide an intuitive plot of  $SO_2$  detection for each volcano in the supplementary material.

---

## Author Response (AR1)

**A multi-sensor satellite-based archive of the largest SO$_2$ volcanic eruptions since 2006, by Pierre-Yves Tournigand et al.**

**Anonymous Referee #1**

We would like to thank the anonymous reviewer for the insightful and constructive comments, which helped us to improve our manuscript. We appreciate the valuable comments and tried to address the issues raised as best as possible.

Specific Comments
MANUSCRIPT
Reviewer: Line 17. I feel it would be beneficial to elaborate here on there being no other archive which compiles the results from multiple satellite instruments and eruptions to really emphasise the strength of this dataset.
See after
Reply: We now emphasized it in different sections of the paper.

Reviewer: Line 20. 'We've archived and collocated … the vertical backscatter from CALIOP …' – I think rather than 'vertical backscatter' this sentence should indicate that you have included the CALIOP height and aerosol type.
Reply: Corrected.

Reviewer: Line 25. Here you state that 'the cross-comparison of the datasets shows the high consistency of the parameters estimated with different sensors and algorithms'. This feels like quite a strong statement. In section 5 you compare the heights obtained with RO, CALIOP and IASI. You note that for a number of eruptions there is a good agreement between RO and CALIOP but that this is not the case for Calbuco. Table 4 shows that a number of the average differences between IASI/CALIOP and IASI/RO are greater than 3 km. Additionally, you have done no quantitative comparison of the partial column densities from AIRS, IASI and GOME-2. Some rewording would improve this statement.
Reply: The current paper introduces a new data archive that combines several satellite data-sets for recent eruptions and, for the first time, includes radio occultation data. Some limited inter-comparisons of the data are already published in the literature (Brenot et al., 2014; Carn et al., 2015; Theys et al., 2013), so here we concentrate on describing the archive. Future papers are planned to inter-compare different estimations of partial column densities and cloud top heights. We reworded the sentence to make it clearer and it now reads:

"*the cross-comparison of the datasets shows different consistency of the parameters estimated with different sensors and algorithms according to the sensitivity and resolution of the instruments*"

- Brenot, H., Theys, N., Clarisse, L., van Geffen, J., van Gent, J., Van Roozendael, M., et al.: Support to Aviation Control Service (SACS): an online service for near-real-time satellite monitoring of volcanic plumes, in: Natural Hazards and Earth System Sciences, 14(5), 1099–1123. https://doi.org/10.5194/nhess-14-1099-2014, 2014.
- Carn, S. A., K. Yang, A. J. Prata, and N. A. Krotkov (2015), Extending the long-term record of volcanic SO2 emissions with the Ozone Mapping and Profiler Suite nadir mapper, Geophys. Res. Lett., 42, doi:10.1002/2014GL062437.
- Theys, N., Campion, R., Clarisse, L., Brenot, H., van Gent, J., Dils, B., Corradini, S., Merucci, L., Coheur, P.-F., Van Roozendael, M., Hurtmans, D., Clerbaux, C., Tait, S., and Ferrucci, F.: Volcanic SO2 fluxes derived from satellite data: a survey using OMI, GOME-2, IASI and MODIS, Atmos. Chem. Phys., 13, 5945–5968, https://doi.org/10.5194/acp-13-5945-2013, 2013.

Reviewer: Line 91-102. This paragraph is a little confusing. You cite three papers/datasets: Ge et al. (2016); Carn et al. (2017); Carn et al. (2019) and it is a little difficult to tell if and how these papers/datasets are connected. Also, you suggest that Carn et al. (2017) included 'passive degassing' and 'main eruptive events' (line 94) but to the best of my knowledge this paper generates long term averaged fluxes that exclude large eruptive events. I would advise some rewording of this paragraph and perhaps some expansion on what is included in the Ge et al. (2016) and Carn et al. (2017) papers which might to help the reader better understand their content.

Reply: The paragraph has been reformulated as follows.

"*Considering $SO_2$ emissions, several datasets and inventories are available and updated over time, but generally include daily or yearly total emissions per volcano or per eruption. Ge et al. (2016) compiled an inventory for daily $SO_2$ emissions in the time frame 2005-2012 including global volcanic eruptions but also eight persistently degassing volcanoes retrieved by the Ozone Monitoring Instrument (OMI) on board the Aura satellite. Carn et al. (2017) implemented it including OMI retrievals from 2005 to 2015 of emissions related to passive degassing. The most updated … provided (Carn et al., 2016; Carn, 2019). The above-mentioned datasets provide important information for users mainly needing to assess the climatic impact of SO2 from volcanic sources, however, none of them allows for mapping the $SO_2$ emissions and related altitude estimations in space and time and thus the direct testing and comparison of new models and techniques, like GNSS RO, for example. We think it is important to provide a complementary multi-satellite archive covering the largest eruptive events and their cloud development all around the world in order to facilitate the access to such data for future studies.*"

Details about Ge et al. (2016) inventory are available at http://wiki.seas.harvard.edu/geos-chem/index.php/Volcanic_SO2_emissions

Reviewer: Section 2. In this section it would be useful to have some more information about the performance of each technique. For example, conditions in which the technique performs well or badly. And information such as the detection limits and uncertainties. This has been done for AIRS (lines 135-138) and something similar for each instrument/technique would help the reader appreciate the strengths and limitations of each tool. It would also help a user to correctly interpret the archived data- especially if they are comparing the results from different instruments. Section 4.3 does point the reader to some of the relevant literature but it would be nice to have this in section 2 and with more detail.

Reply: For IASI limitation and uncertainties please see the next reply.
For GOME limitation and uncertainties we added the following text to the paragraph 2.3:

"*The volcanic emission measurement is facilitated by large $SO_2$ columns generally at high altitudes (free-troposphere to lower stratosphere). However, for large $SO_2$ columns (typically>50 DU) the absorption tends to saturate leading to a general underestimation and directly affecting the product accuracy. For most volcanoes, there is no ground-based equipment to measure $SO_2$ during the eruption and the validation approach is usually a cross-comparisons with other satellite products. The O3M SAF validation report (Theys and Koukouli, 2015) shows that GOME-2 $SO_2$ product reaches the target/optimal accuracy of 50%/30% respectively. It is important to notice that the $SO_2$ retrievals from GOME-2 are also affected by clouds and instrumental noise especially at high solar zenith angles. These limitations have been filtered in the data used in this work, according to the criteria shown by Brenot et al. (2014).*"

- Theys and Koukouli, https://cdop.aeronomie.be/ProjectDir/documents/ValidationReports/Validation_Report_GOME-2_SO2_GDP4.8_Dec2015.pdf
- Brenot, H., Theys, N., Clarisse, L., van Geffen, J., van Gent, J., Van Roozendael, M., et al.: Support to Aviation Control Service (SACS): an online service for near-real-time satellite monitoring of volcanic plumes, in: Natural Hazards and Earth System Sciences, 14(5), 1099–1123. https://doi.org/10.5194/nhess-14-1099-2014, 2014.

Reviewer: Line 144. You mention the IASI retrieval technique is based on a BTD with the v3 absorption band – brightness temperature difference with what?

Line 144-149. Initially it is implied that the IASI VCD retrieval is run using fixed heights. But in the archived data there is only a single value for the IASI VCD. Could you clarify if this is obtained by interpolating the results with the height from the second retrieval?

Line 148. It would be useful if there was a line here explaining how the IASI height retrieval worked.

Reply: Thank you for these comments; we have now updated the relevant paragraph 2.2 clarifying these different aspects (including some general statement on sensitivity and uncertainties):

*"The Infrared Atmospheric Sounding Interferometer (IASI) is a Fourier transform instrument onboard the near-polar sun-synchronous orbiting satellites Metop-A and Metop-B, respectively, launched in October 2006 and September 2012 with ascending equator crossing local time at 9:30. IASI covers the full globe two times per day with a swath of 2200 km and a spatial resolution of 12 km at nadir (Clerbaux et al., 2009). The $SO_2$ retrieval is based on a brightness temperature difference between channels in and outside the $SO_2$ $v_3$ band (Clarisse et al., 2012) which is converted to $SO_2$ concentration integrated along the vertical axis the Vertical Column Density (VCD) using look-up tables and operational profiles of pressure, temperature and humidity. The retrieval of VCD assumes that all $SO_2$ is located at particular atmospheric layers (5, 7, 10, 13, 16, 19, 25 or 30 km above sea level) providing different estimations at different altitudes. It has a detection limit of around 0.5 DU at the tropopause, which increases for decreasing altitude (depending on the amount of water vapour in the atmosphere). For plumes above 500hPa (about 5.5 km) the algorithm has a theoretical uncertainty between 3-6%. A second algorithm (Clarisse et al., 2014) is applied to compute the $SO_2$ cloud altitude with an accuracy of about 2 km for plumes below 20 km. The algorithm exploits the fact that the $SO_2$ $v_3$ band interferes with strong water vapour absorptions, and that these interferences, by virtue of the vertical water vapour profile, have a strong dependency with height. Combining the two datasets, a single best-estimate VCD is obtained by interpolating the VCD columns of the first algorithm at the retrieved height."*

Reviewer: Section 2.4. I think in this section it is important to highlight that with CALIOP you are not measuring SO2 but ash, sulphate, smoke and/or dust. It would be good to acknowledge here some of the limitations of assuming SO2/ash are collocated.

Reply: CALIOP instrument does not allow $SO_2$ measurements but dust, elevated smoke, volcanic ash and sulfate. However, the CALIOP classification algorithm do not include the volcanic ash type below the tropopause level (Kim et al. 2018) so it is difficult to distinguish the volcanic ash from other aerosol types in the lower troposphere. Both ash and $SO_2$ are not necessarily collocated during an eruption, this is the reason why all the CALIOP data have also been collocated with the $SO_2$ estimation from AIRS, IASI and GOME-2. We added a new sentence in the section 2.4 of the manuscript stating:

*"The CALIOP does not allow $SO_2$ measurements or estimation (it provides estimations of dust, elevated smoke, volcanic ash and sulfate) and the CALIOP classification algorithm do not include the volcanic ash type below the tropopause level (Kim et al. 2018) making difficult to distinguish the volcanic ash from other aerosol types in the lower troposphere., For these reasons, the selected CALIOP backscatter is collocated with the $SO_2$ estimation from AIRS, IASI and GOME-2 and this combination provides a complete information on the content and vertical structure of the cloud."*

- Kim, M. H., Omar, A. H., Tackett, J. L., Vaughan, M. A., Winker, D. M., Trepte, C. R., ... & Kar, J. (2018). The CALIPSO version 4 automated aerosol classification and lidar ratio selection algorithm. Atmospheric measurement techniques, 11(11), 6107.

Reviewer: Section 3. This section details all the variables contained in the files. I think it would be really beneficial to a user to have these listed in a table (either in this paper or in the supplementary information). I found I referred to the supplementary information (print out of all the variables) a lot while trying to load and plot the data. A table summarising the variable names, meaning, dimensions, type and units would be even more useful as a quick reference guide.

Reply: We added a table with all the info.

[revised manuscript text omitted]

Reviewer: Line 201. This is the first instance that 'granule' has been used. Please can this be clearly defined. The use of the word granule made it challenging to interpret the data structures described in sections 3.1-3.3 independently of reading in and looking at the data.

Reply: The term "granule" refers to the AIRS data, while, for IASI and GOME-2, we refer to scanning lines. AIRS collects data as it sweeps along the orbit, and the data is then sectioned into pieces called "granules". Each AIRS granule is roughly 2250 x 1650 kilometers and contains 6 minutes of data. There are nominally 240 Level 1B and 240 Level 2 granules of 6-minute duration generated each day. The orbital repeat cycle is 16 days, but orbital maintenance manoeuvres can shift granules along orbits by a small fraction of a granule. Maps showing the locations of granules are generated daily and available for download. AIRS data users use

maps like these when making requests from AIRS data servers. We now explain the term granule in the manuscript with the following sentence:

*"A granule is a portion of AIRS orbit containing 6 minutes (2250 km x 1650 km) of data, which is officially defined by the National Aeronautics and Space Administration (NASA)."*

Section 3.4, Section 3.5. In these sections you mention collocation with between CALIOP, RO and the other instruments. I think it would be useful to know what conditions you use for the collocation here rather than in sections 4.1 and 4.2. This would help the reader immediately understand what is meant by collocation. Reply: Done.

Reviewer: Lines 340-344. Here you discuss the average differences between the cloud heights for different eruptions. Are there any reasons why the average difference is greater at Calbuco than for Eyja, Kasatochi and Grimsvotn? Also, is there a reason why the differences are greater between IASI and RO/CALIOP? I think it would be useful for a user of the dataset to understand why differences might arise between the different datasets (e.g. the time difference between the overpasses and the method used to obtain the height information).
Reply: The reasons are due to different eruption types and the different sensitivity and resolution of the measurement techniques. As reported in section 2.2 (IASI), some assumptions have been made to retrieve the cloud height allowing an estimation with an accuracy of about 2 km. Moreover, the IASI height estimations are sampled every 0.5 km. The RO cloud height estimation is based on the density variation of the atmosphere, so denser clouds (e.g. Kasatochi 2008) can be detected more likely than less dense clouds (e.g. Calbuco 2015) and with better accuracy. Most importantly, the RO and CALIOP are limb profiling techniques with high vertical resolution, while IASI is a nadir sounding technique. This does not allow IASI to provide the same vertical resolution and accuracy that we can get from RO/CALIOP.
In addition, the number of colocations between RO and CALIOP is much smaller than for RO-IASI and IASI-CALIOP, respectively. We revised the text at the end of section 5 (Data cross-comparisons) and added further explanations:

*"The difference in cloud top estimations can be partly explained by the different sensitivities and vertical resolution of the different instruments. In addition, the number of colocations between RO and CALIOP is much smaller than for RO-IASI and IASI-CALIOP, respectively. The cloud top height estimation for eruptions with a large number of colocations (Calbuco, Kasatochi, Nabro and Sarychev Peak) is in general consistent within the techniques."*

Reviewer: Additionally, have you considered a quantitative comparison of the VCDs retrieved with AIRS, IASI and GOME-2? What differences would you expect to see between these?
Reply: As we reported above, the current paper introduces a new data archive that combines several satellite data-sets for recent eruptions and, for the first time, includes radio occultation data. Some limited inter-comparisons of the data are already published in the literature (Brenot et al., 2014; Carn et al., 2015; Theys et al., 2013), so here we concentrate on describing the archive. In Table 3 we report the $SO_2$ mass loading for each eruption with different instruments reported in literature. We prefer to refer to published studies, instead of re-computing the mass loadings in these specific cases, to avoid confusion to the readers.

Reviewer: Table 1. You could add the eruption VEI and the eruption end date or duration to this table. Additionally, it could be helpful to add the geographic region considered for each eruption and the start/end date for the data in the archive – both of these would be valuable to the data user.
Reply: We added to the table the VEI and the archive start/end dates for each eruption. Please note that the VEI is not always appropriate for $SO_2$-rich eruptions since it corresponds to ash-rich eruptions. Instead of adding the geographic region in the table, we prefer to provide an intuitive plot of $SO_2$ detection for each volcano in the supplementary material.

[Figure]

Figure S1. Okmok cloud map.

[Figure]

Figure S2. Kasatochi cloud map.

[Figure]

Figure S3. Sarychev cloud map.

[Figure]

.
Figure S4. Eyjafjallajokull cloud map.

[Figure]

Figure S5. Merapi cloud map.

[Figure]

Figure S6. Grimsvotn cloud map.

[Figure]

Figure S7. Nabro cloud map.

[Figure]

Figure S8. Puyehue Cordon Caulle cloud map.

[Figure]

Figure S9. Tolbachik cloud map.

[Figure]

Figure S10. Kelut cloud map.

[Figure]

Figure S11. Calbuco cloud map

Reviewer: Table 2. In addition to the information given you could also mention the spectral range/resolution of the instruments.
Reply: Table 2 was modified accordingly.

Reviewer: Figure 2. It would be interesting to see the cloud top heights obtained with CALIOP and RO in this plot rather than just the tracks/points.
Reply: Corrected. We added the average values of cloud top heights in each panel. We believe that reporting the values (with numbers or with different colors) on these maps, could make the figure difficult to read. We also added a short discussion in the new section "Results" explaining how the archive can be used to compare the cloud top heights computed with different instruments (Tournigand et al., 2020).

- Tournigand, P.-Y., Cigala, V., Prata, F., Steiner, A. K., Kirchengast, G., Brenot, H., Clarisse, L., Biondi, R.: The 2015 Calbuco volcanic cloud detection using GNSS radio occultation and satellite lidar, IGARSS 2020 Proceedings, accepted.

SUPPLEMENT AND DATA
Reviewer: NULL values – Throughout much of the dataset the null values are reported as -9999. However, for the RO profiles they are recorded as NaN. For the RO cloud top heights it goes back to -9999. For the CALIOP heights there are no -9999 or NaN instead there are 0's- are these null values too? This should probably be consistent and whichever is chosen should be clearly noted somewhere.
Reply: the archive has been modified, we decided to use -9999 as common filling value.

Reviewer: Different number of variables - The files do not contain consistent numbers of variables. For example, in the file 'Calbuco_2015_05_24.nc' there is data available for IASI and RO but not GOME-2, AIRS or CALIOP. Presumably this is related to the availability of the data. It would be good to clarify this in the manuscript (perhaps at the start of section 3). Even better would be to summarise how many days or which

days are covered by each instrument for each eruption – this could be an addition to table 3 and would be slightly easier to interpret than the number of granules.

Reply: Yes, just the available instruments are reported in this archive. We have updated the text at Line 199:

*"…the variables available from one day to another may differ according to SO₂ detection results and instruments availability"*.

As suggested by the reviewer, we have also added to the table 3 the number of days covered by each instrument for each eruption.

Reviewer: Dimensions - a list of dimensions is given on page 1 and page 8 of the supplement. It would be very helpful if these were expanded on. In particular the definitions for 'CALIOP_char' and 'CALIOP_char2' are not very informative.

Reply: Dimensions descriptions have been expanded.

Reviewer: P1, P2 supplementary info – there is a slight discrepancy between the long names between IASI and GOME-2. For GOME-2 the long name states that the data is a composite of GOME-A and B – is it also the case for IASI that the data is a composite of IASI-A and B?

Reply: the archive and supplementary information have been modified accordingly.

IASI_SO2 – I suggest expanding the long name of this variable to make it clear that this is a vertical column density and to explain what interpolated is referring to P3, P4. It is not clear what the dimensions should be here.

Reply: the archive and supplementary information have been modified accordingly.

Reviewer: CALIOP_CHAR, CALIOP_char2 and CALIOP_type should be more clearly defined in the supplementary data. CALIOP_type (the dimension) is not defined in the dimensions list.

Reply: Corrected.

Reviewer: CALIOP_type – This variable was very challenging for me to read in (in both IDL and python). The supplementary information (page 4) suggested that these were doubles but they had to be read in as strings. I think the choice of saving these as a string is so that multiple flags can be indicated. Initially on reading in this variable I obtained an array with 3 dimensions. These then had to be converted to strings and joined together to extract the CALIOP type (a similar thing had to be done for CALIOP_filename- also not immediately obvious how to read in IDL). Following that the newly joined strings had to be searched to determine which aerosols were present. Could there be a better way of saving this variable? Perhaps simply an integer array for each variable type with 1 indicating the presence of this aerosol and 0 indicating its absence. Alternatively, more information on how to read in and interpret these results would be very useful.

Reply: Indeed, this variable was indicated as double while it is a string. This was corrected (see section 3.4). It is also correct that the choice of saving these data as string is to allow multiple flags. We didn't elect to use an integer array of 0 and 1 because we think that the possibility to distinguish one aerosol type from another is crucial for the user of the archive. For example, the user will be able to know if ash is likely to be present in the area of interest together with the SO₂ detected by AIRS, IASI and GOME-2. Finally, the description of the variable's dimensions has been modified in order to allow the user to better understand how to use it.

Reviewer: P5-P7. For the RO variables expanding the long names for 'air_temperature', 'air_pressure', 'refractivity', 'specific_humidity' would provide more information- these could for example mention that these are profiles.

Reply: the archive and supplementary information have been modified accordingly.

Reviewer: RO – cloud top heights. The units do not seem to be consistent for these (in the daily files). For colocations with AIRS and IASI the heights appear to be in meters (which are the standard units and consistent with heights reported by CALOP and IASI). Whereas for GOME-2 they seem to be in km.

Reply: the archive has been modified accordingly.

Reviewer: P4-7. The dimensions for the RO profiles are listed as RO_AIRS_lat by RO_AIRS_PROFILE (or IASI/GOME). Could these be defined more clearly in the dimensions list.
Reply: Corrected.

Reviewer: Dates covered by each eruption. Some of the daily files start before the start date of the eruption. For example, for Nabro (eruption starting on the 13th June 2011) the first file in the dataset is 31st May 2011. In the first few files it seems to include the outputs for other eruptions. For example, the file Nabro_2011_05_31 includes SO2 measurements from the Grímsvötn eruption, while the file Nabro_2011_06_05 includes measurements from both Grímsvötn and Puyehue. Including this twice in the dataset is a little unnecessary and means the user has to download more data than is needed for this eruption. It is possible to see plumes from different eruptions in many of the datafiles.
Reply: the archive has been modified accordingly.

Technical Comments/Suggestions
MANUSCRIPT
Reviewer: Throughout – Some of the volcano names have accents (e.g. Grímsvötn, Eyjafjallajökull, Puyehue-Cordón Caulle)
Reply: Corrected.

Reviewer: Line 16. 'Forecast' should be forecasting or forecasts
Reply: Corrected.

Reviewer: Line 17. 'Single events' would be more precise as 'single eruptive events'
Reply: Corrected.

Reviewer: Line 17. '… but not any archive is available' need rewording. Perhaps: '… no such archive is available'.
Reply: Corrected.

Reviewer: Line 18. 'from three different instruments' would be clearer as 'from three different satellite instruments'
Reply: Corrected.

Reviewer: Line 19. 'the atmospheric parameters vertical profiles from …' This line is a little confusing.
Reviewer: Perhaps rephrasing as something like: 'vertical atmospheric profiles obtained from …'
Reply: Corrected.

Reviewer: Line 21. 'We additionally' would read better as 'Additionally we'
Reply: Corrected.

Reviewer: Line 22. 'The dataset consists of 223 days monitored with SO2 clouds' This line does not read very well – consider rephrasing it.
Reply: Corrected.

Reviewer: Line 38-39. What is meant by 'consequent cloud'? – are you referring to the volcanic cloud or ice/water clouds (e.g. indirect climate effects)
Reply: Corrected.

Reviewer: Line 40. 'SO2 injections in the stratosphere' may read better as 'SO2 injections into the stratosphere'
Reply: Corrected.

Reviewer: Line 42. 'hence transported' may read better as 'hence be transported'
Reply: Corrected.

Reviewer: Line 46. 'has occurred per year since 1994 worldwide' might read better as 'have occurred worldwide each year since 1994'
Reply: Corrected.

Reviewer: Line 47-48. '… the energy of the eruption, amount, type and size of the ejected material' would read better as '… the energy of the eruption, and the amount, type and size of the ejected material'
Reply: Corrected.

Reviewer: Line 49-51. To improve sentence clarity move the Newhall and Self reference to the start of the sentence: 'The VEI was introduced in 1982 by Newhall and Self (1982) …'
Reply: Corrected.

Reviewer: Line 50. I think it is Richter scale rather than Richter's scale.
Reply: Corrected.

Reviewer: Line 50. I think it should be earthquake rather than earthquakes'
Reply: Corrected.

Reviewer: Line 54. 'VEI index' can just be VEI
Reply: Corrected.

Reviewer: Line 60. Putting 'e.g. VEI 4 events' within brackets would help the readability of the Sentence
Reply: Corrected.

Reviewer: Line 71. 'and' should be used instead of 'or'
Reply: Corrected.

Reviewer: Line 72. 'although' would make more sense than 'even though'
Reply: Corrected.

Reviewer: Line 74. 'focusing on single or a few eruptions' would read better as 'focusing on a single or a few eruptions'
Reply: Corrected.

Reviewer: Line 77. Stating that 'all' platforms and algorithms were studied in this volume seems quite strong. Perhaps: 'a large number' would be better
Reply: Corrected.

Reviewer: Lines 77 and 81. Starting the sentence with 'Sarychev Peak 2009' and 'Grimsvotn 2011' does not read very well. It might sound better as 'The Sarychev Peak eruption in 2009 …' etc.
Reply: Corrected.

Reviewer: Line 91. '… and updated in the course of the years.' This line does not read very well – consider rephrasing.
Reply: Corrected.

Reviewer: Line 110. It should read '... and humidity from GNSS RO profiles'

Reply: Corrected.

Reviewer: Line 111-112. This sentence would benefit from being rewritten to improve the clarity. Maybe something like: 'This information is provided for eruptions, after 2006, classified by the GVP as VEI 4 or larger and with an SO2 mass loading of greater than 0.05 Tg. At the time of archive preparation, no eruptions after 2016 had yet been classified as VEI 4 or greater.'
Reply: Corrected.

Reviewer: Line 113. Rather than include '(table 1)' in this sentence, perhaps add a sentence at the end of the paragraph saying 'Further information on these eruptions can be found in table 1.'
Reply: Corrected.

Reviewer: Line 117-118. 'there is no current unique database'. This does not read very well – I would suggest rewriting the sentence
Reply: Corrected.

Reviewer: Line 119-121. It should read 'accurate knowledge of volcanic SO2 cloud concentration and altitude as well as their spatial and temporal evolution: : : of an eruption's climatic impact'
Reply: Corrected.

Reviewer: Line 122. 'retrievals' should be 'retrieval'
Reply: Corrected.

Reviewer: Section 2 – title. Maybe this should be titled 'Instrument and Retrieval Description'
Reply: Corrected.

Reviewer: Line 126. 'due to their own limitations.' It is not clear what is meant by this.
Reply: Corrected.

Reviewer: Line 130. It should read 'an ascending orbit'
Reply: Corrected.

Reviewer: Line 133. It would be good for a reference to be included for this sentence so the reader is immediately aware of which paper describes this technique.
Line 135. Again it would be good to have a reference for this statement.
Reply: We now added a sentence to text stating

"*The AIRS SO$_2$ retrieval used is described in detail by Prata and Bernardo (2007); here we provide a very brief overview.*"

Reviewer: Line 142. It should read 'an ascending orbit'
Reply: Corrected.

Reviewer: Line 144. V3 has not been defined. SO2 is also not formatted correctly.
Reply: Corrected.

Reviewer: Line 151. Slight inconsistency - On board has a space here but elsewhere it is written onboard.
Reply: Corrected.

Reviewer: Line 153. Slight inconsistency - Here the pixel size is listed as 40x80 km. For AIRS it was written as 13.5 x 13.5 km (with spaces).
Reply: Corrected.

Reviewer: Line 165. 1,67 should be 1.67
Reply: Corrected.

Reviewer: Line 175-176. It may read better as 'In this archive we use the RO bending …'
Reply: Corrected.

Reviewer: Line 192. It should read 'the number of days' rather than 'amount of days'
Reply: Corrected.

Reviewer: Line 232-233: 'Four of those types are of interest for this archive: type 2, 6, 9 and 10 respectively corresponding to dust, elevated smoke, volcanic ash and sulfate/other.' Include a colon between archive and type.
Reply: Corrected.

Reviewer: Line 234-235. There should be a space between 20.2/30.1 and km
Reply: Corrected.

Reviewer: Line 240. I think a colon would be better than a comma between provided and latitude
Reply: Corrected.

Reviewer: Line 264. Should read 'Where _ is the bending angle anomaly …'
Reply: Corrected.

Reviewer: Line 277. Should read 'consists of' rather than 'consists in'
Reply: Corrected.

Reviewer: Line 285. I think this should be 'Thus we' rather than 'We thus'
Reply: Corrected.

Reviewer: Line 308. Having a list (in brackets) of parameters that affect the uncertainty, followed by a line about altitude affecting the uncertainty does not read so well. Maybe combine all the factors that affect the uncertainty into one line.
Reply: Corrected.

Reviewer: Line 315. Should read 'volcanic cloud' detection rather than 'volcanic clouds'
Reply: Corrected.

Reviewer: Line 317. It should read 'in charge of processing them'
Reply: Corrected.

Reviewer: Line 317. Is it not 10 VEI 4 and 1 VEI 5 eruptions? The Puyehue eruption in 2011 is listed on the GVP as VEI 5 (https://volcano.si.edu/volcano.cfm?vn=357150 ; under eruptive history). Also here you state the period you a looking at is 2008 to 2016 when previously you've said you were looking at eruptions from 2006-2016.
Reply: Corrected.

Reviewer: Line 318. 'With a total of' rather than 'for a total of'
Reply: Corrected.

Reviewer: Line 320-321. 'Several parameters are measured using different instruments, such as SO2 VCD and cloud top altitude, to allow cross correlation between the different retrieval algorithms.' Do you mean to say

'Several parameters are included within the dataset: : : to allow cross correlation between the different algorithms' ?
Reply: Corrected.

Reviewer: Line 335. It should read 'compared the date, time …'
Reply: Corrected.

Reviewer: Line 336. 'We have additionally' would read better as 'Additionally we have'
Reply: Corrected.

Reviewer: Line 360. Not just detection but also the retrieval of VCDs
Reply: Corrected.

Reviewer: Line 365. 'Up to date' would read better as 'At present'
Reply: Corrected.

Reviewer: Line 370. 'and test new algorithms contributing to improving the accuracy on the estimation of fundamental volcanic clouds parameters'. This may read better as 'and test new algorithms on, thereby contributing to improving the accuracy on the estimation of fundamental volcanic clouds parameters'
Reply: Corrected.

Reviewer: Line 373. 'allowing to reconstruct …' may sound better as 'allowing the reconstruction of …'
Reply: Corrected.

Reviewer: Table 2. Maybe differentiate between AIRS and IASI spatial resolutions (13.5 by 13.5 km vs. 12 km diameter circular pixels)
Reply: Corrected

Reviewer: Figure 2. In the caption 'upright' should be 'top right'. Also, this caption reads a little strangely. I would suggest: 'Example of data use and data collocation. (a) Kasatochi cloud on 9th August 2008; (b) Sarychev peak cloud …' At present there is no (b). Additionally, no full stop is required in line 610.
Reply: Corrected.

SUPPLEMENT AND DATA
Reviewer: P1. VC – is undefined in the supplementary material.
Reply: Corrected.

Reviewer: P1. DATE_IASI – The use of the word 'because' in this description doesn't make sense.
Reply: Corrected.

Reviewer: P2, P9. In the GOME_lon variable- dimensions include GOME_late rather than GOME_lat
Reply: Corrected.

**Anonymous Referee #2**

We would like to thank the anonymous reviewer for the insightful and constructive comments, which helped us to improve our manuscript. We appreciate the valuable comments and tried to address the issues raised as best as possible.

Reviewer Comments
Manuscript ESSD-2020-109, Tournigand et al., A multi-sensor satellite-based archive of the largest SO2 volcanic eruptions since 2006

The authors describe a new data archive that combines synoptic maps of SO2 plumes, derived from UV and TIR radiance measurements, with estimates of plume heights and thickness derived from lidar profiles and radio occultation (RO). This archive will prove to be of great value to studies of gas emissions for the selected eruptions, as accurate knowledge of the height and thickness of plumes is critical to the estimation of gas concentrations. The authors have done a great service by collocating the RO products with the UV, TIR, and lidar data products, thus facilitating the incorporation of RO products into future analyses of the combined data record.

Reviewer: Unfortunately, this manuscript falls short as a description of the new data archive and show-case for the potential applications of the combined data records. With few exceptions, the descriptions of the data products are too sparse to be of much use to anyone but the most experienced remote sensing specialists. The exceptions to the sparse descriptions are focused on the RO products and, to a lesser extent, lidar profiles. Despite the attention paid to the RO data processing, the authors fail to discuss the basic relationships between the RO signals and volcanic plume phenomena. For example, what are the bending angles and bending angle anomalies (Sections 3.5, 4.1.2, and 4.1.3) and what do these anomalies tell us about volcanic plumes? Similarly, what is the relation between the RO refractivity and plume heights?
Reply: GNSS RO is an active limb sounding technique which uses the signals transmitted by a GNSS satellite and received by a Low Earth Orbit satellite, where the atmosphere is vertically scanned due to the relative motion of the two satellites. A ray crossing the atmosphere is refracted, i.e. bent, according to Snell's law due to the vertical gradient of atmospheric density. The effect of the atmosphere is represented by a bending angle, from which refractivity and density is retrieved. Refractivity in the neutral atmosphere depends mainly on temperature, pressure, and water vapour pressure. In other words, the bending angle is a function of the density of the atmosphere which depends on temperature, pressure and humidity.
Depending on the type of volcanic eruptions, the presence of volcanic clouds can modify the vertical structure of the atmosphere in different ways. Some eruptions eject large amounts of ash and $SO_2$ affecting the density of the atmosphere in the region of the cloud, some other eruptions are rich in water vapour. Some eruptions, specifically explosive volcanic eruptions, can also impact atmospheric temperature due to radiative effects. The vertical bending angle anomaly profile shows the perturbation given by the presence of the volcanic cloud since it is computed by subtracting the climatological bending angle profile of the area from the actual bending angle profile co-located with the cloud. The anomaly is thus associated with the perturbation that the volcanic cloud produces in the atmosphere. The largest discontinuities in the vertical bending angle anomaly profile are evident at the height corresponding to the volcanic cloud top. We made this clearer in the revised manuscript text at the beginning of section 2.5:

*"The Global Navigation Satellite Systems (GNSS) Radio Occultation (RO) is an active limb sounding technique which uses radio signals transmitted by a GNSS satellite and received by a Low Earth Orbit satellite, where the atmosphere is vertically scanned due to the relative motion of the two satellites. The signal, travelling through the atmosphere, is refracted and bent due to the vertical gradient of atmospheric density. The effect of the atmosphere is represented by a bending angle, from which refractivity and density is retrieved. Refractivity in the neutral atmosphere depends mainly on temperature, pressure, and water vapour pressure. Information about the vertical structure of the troposphere and stratosphere is provided (Kursinski et al., 1997). ….*

*We also provide the bending angle anomaly which is proven to be an efficient parameter to reveal the impact of the VC on the atmospheric structure (Biondi et al., 2017; Cigala et al., 2019; Stocker et al., 2019) because perturbations in the vertical structure are seen in the bending angle profile as anomalous peaks, specifically at the volcanic cloud top. …"*

Reviewer: The processing of the CALIOP (lidar) data is discussed in the context of the RO data (Section 4.2.1), and the shortcomings outlined above affect the CALIOP discussion as well. The procedure for removing "unnecessary" information from the CALIOP images is confusing. How do the authors determine the "zones of interest" upon which to focus their processing? The practice of discarding CALIOP data corresponding to altitudes below 10 km seems short-sighted. If the RO data products are noisy for such altitudes, then shouldn't the lidar data for low altitude (< 10 km) clouds be all the more important? Since the authors never explain the significance of the RO bending anomaly, relative to volcanic plume heights, it is difficult to appreciate why the low-altitude lidar data should be discarded.

Reply: The zone of interest in CALIOP image is determined using the latitude and longitude of the collocated RO profile and by selecting a window around those values as explained line 292. This part of the text may have been unclear and we rephrased it as:

*"The first step of the cloud top detection procedure consists of cropping the CALIOP backscatter image according to the collocated RO profile position at +/-14° in latitude and +/-80° in longitude. The objective of this first step is to focus the processing on a restricted zone around the position of the collocated RO profile in order to save computational time"*.

The CALIOP data are not provided within this archive due to the large file dimension and the quick and easy way to download them from the NASA portal. What we provide here is the CALIOP filename allowing to quickly retrieve the original data within which it is possible to get all the different parameters including the total attenuated backscatter (at 532 and 1064 nm) and depolarization ratio. However, we provide the complete RO profile (from the surface to 60 km). We thus never provide in this archive data arbitrarily incomplete.

We only cropped the CALIOP backscatter images at 10 km during the procedure of cloud top determination. This choice was made for several reasons. First, all the volcanic clouds in this study reached a maximum altitude higher than 10 km. Second, below 10 km the water vapour content and the presence of meteorological clouds is much more pronounced and it tends to disturb the cloud top identification. The confusion may have come from unclear explanations, we thus rephrased them. Again, this selection is just part of the cloud top detection algorithm, but the full RO profile is provided in the archive and the CALIOP information is kept also for the cases for which we do not detect any cloud top with the RO. Last but not least, CALIOP classification algorithm do not include the volcanic ash type below the tropopause level (Kim et al. 2018) so it is difficult to distinguish the volcanic ash from other aerosol types in the lower troposphere. The manuscript has been now updated as:

*"Below 10 km altitude, the RO bending angle anomaly is noisy due to the presence of moisture. Thus we only included in this archive volcanic clouds with a top altitude above 10 km. The following step in the volcanic cloud top determination is then to remove image information below 10 km which also removes a significant part of meteorological clouds increasing the volcanic cloud top altitude detection accuracy"*.

We also added a new sentence in the section 2.4 of the manuscript stating

*"The CALIOP does not allow $SO_2$ measurements or estimation (it provides estimations of dust, elevated smoke, volcanic ash and sulfate), however, the selected CALIOP backscatter is collocated with the $SO_2$ estimation from AIRS, IASI and GOME-2. Moreover, The CALIOP classification algorithm do not include the volcanic ash type below the tropopause level (Kim et al. 2018) making difficult to distinguish the volcanic ash from other aerosol types in the lower troposphere."*

- Kim, M. H., Omar, A. H., Tackett, J. L., Vaughan, M. A., Winker, D. M., Trepte, C. R., ... & Kar, J. (2018). The CALIPSO version 4 automated aerosol classification and lidar ratio selection algorithm. Atmospheric measurement techniques, 11(11), 6107.

Reviewer: Similarly, the cloud aspect ratio is not explained, and the threshold value of 0.09 is not justified. What is the significance of a low aspect ratio? What are typical aspect ratios of volcanic plumes vs. meteorological clouds? Finally, the authors do not include CALIOP depolarization ratios in the archive. The CALIOP Aerosol Type (included in the Archive) does not identify volcanic ash or sulfates uniquely, while the depolarization ratios document variations in the size and aspect of particles within volcanic plumes.

Reply: Low aspect ratio means a thin cloud. Analysing our dataset, within all the RO timely collocated with CALIOP backscatter (with the constraints explained in the previous point), it turns out that volcanic clouds are thinner than meteorological clouds. We have computed the aspect ratio for all the clouds (defined as volcanic by CALIOP) exactly collocated in space with the ROs and in a strict time range of 2 hours finding out that all the clouds within these constraints show an aspect ratio lower than 0.09. On the other side, the meteorological clouds usually show an aspect ratio higher than 0.1 except for a few cases. At the end of section 4.2.1 we now added the sentence

*"Based on all the collocations between CALIOP and the GNSS RO, a statistical analysis of volcanic clouds defined by the CALIOP cloud mask and collocated with the RO within 2 hours, has shown that the volcanic clouds are usually thinner than meteorological clouds with an aspect ratio lower than 0.09. According to this result, the final stage of the algorithm consists of distinguishing clusters corresponding to volcanic features from the ones corresponding to meteorological clouds setting the higher limit of the aspect ratio to 0.09. Finally, the remaining clusters' top altitudes are measured and an average value calculated and saved in the archive as an estimate of the volcanic cloud top altitude."*

As also reported in the previous reply, the CALIOP data are not provided within this archive due to the large file dimension and the quick and easy way to download them from the NASA portal. What we provide here is the CALIOP filename allowing to quickly retrieve the original data within which it is possible to get all the different parameters including the total attenuated backscatter (at 532 and 1064 nm) and depolarization ratio.

Reviewer: In an effort to justify the creation of the archive, the authors make a number of problematic statements. The statement that "… not any archive is available at the moment to be used as background for future studies" (Line 17) is not true, as the authors demonstrate by citing the existing LaMEVE (Lines 68-70), OMI (Lines 92-93), and TOMS/OMI MSVOLSO2L4 (Lines 95-96) archives. In addition, the authors acknowledge the GVN archive at many locations in the text. The new archive may be the first to include RO data, but the potential contributions of RO to volcanology (see my previous comments) have never been discussed. Consequently, the unique nature of the new archive is not obvious.

Reply: It is true that other archives collecting data about volcanic ash clouds and SO2 clouds exists. However, these archives are focusing on one volcano and/or data from one instrument and/or on ancillary information (e.g. LaMEVE). This archive brings together quantitative data on volcanic clouds for the first time from several major eruptions observed through several instruments and for the very first time including GNSS RO data. We agree with reviewer 2 on the fact that our statement was not phrased correctly and would have been potentially confusing for the reader. We modified the text accordingly:

Line 16: *"Several papers have been published focusing on single eruptive events, but no archive available at the moment combines quantitative data from as many instruments"*.

Line 75: *"although these types of database are generally limited to ancillary information, a specific volcano, a specific time window or a specific instrument (e.g. de Moor et al., 2017)"*.

Reviewer: The authors claim that papers describing individual eruption events "make it difficult to compare" the data sets (Lines 74-75) is disingenuous. The authors neglect to mention that the studies cited in this

paragraph (Lines 74-90) The authors present no evidence for the claim that the volcanic clouds generated by the Merapi, Tolbachik, Kelut, and Calbuco eruptions were not studied "in depth" (Lines 87 – 90) What is the definition of "in depth?" The Calbuso eruption clouds, in particular, have been studied extensively, as this eruption had an impact of the evolution of the southern ozone hole.

Reply: The first statement line 74-75 was indeed potentially too strong and actually not really necessary. We elected to remove it. The term "in depth" was probably not appropriate. We choose to rephrase this sentence:

*"The Calbuco 2015 eruption (Marzano et al., 2018; Lopes et al., 2019) was widely studied especially in connection to its impact on the Antartic ozone hole (Ivy et al., 2017; Stone et al., 2017; Zhu et al., 2018; Zuev et al., 2018). The rest of the volcanic clouds, such as the ones produced by Merapi 2010 (Picquout et al., 2013), Tolbachik 2012 (Telling et al., 2015), and Kelut 2014 (Kristiansen et al., 2015; Vernier et al., 2016) also received some attention from the scientific community"*.

- Ivy, D. J., Solomon, S., Kinnison, D., Mills, M. J., Schmidt, A., & Neely, R. R. (2017). The influence of the Calbuco eruption on the 2015 Antarctic ozone hole in a fully coupled chemistry-climate model. Geophysical Research Letters, 44(5), 2556-2561.
- Stone, K. A., Solomon, S., Kinnison, D. E., Pitts, M. C., Poole, L. R., Mills, M. J., ... & Vernier, J. P. (2017). Observing the impact of Calbuco volcanic aerosols on South Polar ozone depletion in 2015. Journal of Geophysical Research: Atmospheres, 122(21), 11-862.
- Zhu, Y., Toon, O. B., Kinnison, D., Harvey, V. L., Mills, M. J., Bardeen, C. G., ... & Jégou, F. (2018). Stratospheric aerosols, polar stratospheric clouds, and polar ozone depletion after the Mount Calbuco eruption in 2015. Journal of Geophysical Research: Atmospheres, 123(21), 12-308.
- Zuev, V. V., Savelieva, E. S., & Parezheva, T. V. (2018). Study of the Possible Impact of the Calbuco Volcano Eruption on the Abnormal Destruction of Stratospheric Ozone over the Antarctic in Spring 2015. Atmospheric and Oceanic Optics, 31(6), 665-669.

Reviewer: The authors need to include at least one example of unique contribution of the new archive to plume studies. Figure 2 shows the locations of data points, but not the unique contributions to the study of plumes enabled by the new archive. For example, what are the levels of agreement between the UV- and TIR-based SO2 retrievals? What are the variations in SO2 retrievals relative to atmospheric conditions (principally temperature and humidity) and plume altitude? What are the levels of agreement between the IASI, CALIOP, and RO-based plume altitude estimates? Which altitude estimates have the highest level of confidence? Examples of the potential contribution of the new archive to plume studies would help with the troublesome justifications for the new archive. The contributions would become readily apparent, eliminating the need for the current unsupported statements.

Reply: The current paper introduces a new data archive that combines several satellite data-sets for recent eruptions and, for the first time, includes radio occultation data. Some limited inter-comparisons of the data are already published in the literature (Brenot et al., 2014; Carn et al., 2015; Theys et al., 2013), so here we concentrate on describing the archive. A future paper is planned that demonstrates how to use the data for some specific cases. The validation of $SO_2$ from GOME-2 and IASI is also part of the EUMETSAT Satellite Application Facility on Atmospheric Composition and Monitoring (AC SAF) activity. The objective of this paper is to present the organisation of an archive grouping data from different instruments, but not to discuss the efficiency of each retrieval algorithm used by each instrument.

Table1 shows the agreement between IASI, CALIOP and RO-based plume altitudes. The average discrepancies between IASI and CALIOP, IASI and RO and CALIOP and RO are respectively 2.0 km, 1.7 km and 0.8 km. Consequently, we have the lowest confidence in IASI data.

We agree with the reviewer 2 that some examples of unique contribution of the archive is needed. So we added a new section to the manuscript titled "Results", where we summarize the content of the archive and we show with some examples (and citations to previous works) how the dataset can be used. Also a new figure (related to the section Results) has been added to the manuscript. the Figure 3 shows an example of dataset usefulness: The archive allows the user to collocate the different sensors (maps on the right), to check the vertical structure of the cloud (aerosol types from CALIOP), to analyse the effect that the cloud has in the

atmospheric structure in terms of density (bending angle anomaly), humidity and temperature and to compare the different cloud top estimations.

[Figure]

Figure 3. RO profiles corresponding to the CALIOP aerosol types profile co-located with the SO$_2$ estimation from IASI (top panels, 12/08/2008) and AIRS (bottom panel, 11/08/2008).

- Brenot, H., Theys, N., Clarisse, L., van Geffen, J., van Gent, J., Van Roozendael, M., et al.: Support to Aviation Control Service (SACS): an online service for near-real-time satellite monitoring of volcanic plumes, in: Natural Hazards and Earth System Sciences, 14(5), 1099–1123. https://doi.org/10.5194/nhess-14-1099-2014, 2014.
- Carn, S. A., K. Yang, A. J. Prata, and N. A. Krotkov (2015), Extending the long-term record of volcanic SO2 emissions with the Ozone Mapping and Profiler Suite nadir mapper, Geophys. Res. Lett., 42, doi:10.1002/2014GL062437.

[revised manuscript text omitted]

735      **Figure 1. Archive schematic organization.**

[Figure]

[Figure]

740 **Figure 2. Example of data use and data collocation. (a) Kasatochi cloud on 9th of August 2008 ; (b) Sarychev Peak cloud on 12th of June 2009. The central panels show the SO₂ VCD 2-dimensional spreading estimated by AIRS, GOME-2 and IASI, the  top right panel show the CALIOP tracks for which the total attenuated backscatter profile is available and the bottom-left panel shows the RO profiles collocated with the SO₂. For IASI, CALIOP and RO the average volcanic cloud top altitude for the considered day is indicated (VC h).**

[Figure]

745

**Figure 3. Case study Kastochi 2008. RO profiles corresponding to the CALIOP aerosol types profile collocated with the SO$_2$ estimation from IASI (top panels, 12/08/2008) and AIRS (bottom panels, 11/08/2008).**